**Registered Report**

# Effects of calibrated blue–yellow changes in light on the human circadian clock

Christine Blume [1,2,3] ✉, Christian Cajochen [1,2], Isabel Schöllhorn [1,2], Helen C. Slawik[4] & Manuel Spitschan [5,6,7] ✉

Evening exposure to short-wavelength light can affect the circadian clock, sleep and alertness. Intrinsically photosensitive retinal ganglion cells expressing melanopsin are thought to be the primary drivers of these effects. Whether colour-sensitive cones also contribute is unclear. Here, using calibrated silent-substitution changes in light colour along the blue–yellow axis, we investigated whether mechanisms of colour vision affect the human circadian system and sleep. In a 32.5-h repeated within-subjects protocol, 16 healthy participants were exposed to three different light scenarios for 1 h starting 30 min after habitual bedtime: baseline control condition (93.5 photopic lux), intermittently flickering (1 Hz, 30 s on–off) yellow-bright light (123.5 photopic lux) and intermittently flickering blue-dim light (67.0 photopic lux), all calibrated to have equal melanopsin excitation. We did not find conclusive evidence for differences between the three lighting conditions regarding circadian melatonin phase delays, melatonin suppression, subjective sleepiness, psychomotor vigilance or sleep. The Stage 1 protocol for this Registered Report was accepted in principle on 9 September 2020. The protocol, as accepted by the journal, can be found at https://doi.org/10.6084/m9.figshare.13050215.v1.

## Effects of light on circadian physiology

Light profoundly affects human physiology and behaviour via the retinohypothalamic (RHT) pathway that relays information from the retina to the circadian pacemaker in the suprachiasmatic nuclei[1]. In humans, this pathway has been functionally mapped out by examining acute responses to evening or night-time light exposure on melatonin secretion, or its circadian phase-shifting effects. Both neuroendocrine (that is, evening and nocturnal melatonin suppression[2–4]) and circadian-phase shifting responses[5,6] to light are primarily driven by the photopigment melanopsin expressed in subset of so-called intrinsically photosensitive retinal ganglion cells (ipRGCs)[7].

## Inputs to circadian phototransduction

However, melanopsin-expressing ipRGCs also receive input from the classical retinal photoreceptors, the cones (active during daylight light levels) and rods (sensitive to dim light but saturated in daylight). Primate ipRGCs receive excitatory synaptic input from the L and M cones and the rods, and inhibitory inputs from the S cones[8] mediated by an S-cone amacrine cell[9] (Fig. 1a). In humans, evidence for this inhibitory input to ipRGCs has been found in the pupillary light response, where S-cone input inhibits the melanopsin-induced pupil constriction[10,11]. A recent direct test for an S-cone involvement in melatonin suppression in the evening has led to the conclusion that they do not contribute

[1]Centre for Chronobiology, Psychiatric Hospital of the University of Basel, Basel, Switzerland. [2]Research Cluster Molecular and Cognitive Neurosciences, University of Basel, Basel, Switzerland. [3]Department of Biomedicine, University of Basel, Basel, Switzerland. [4]Psychiatric Hospital of the University of Basel, Basel, Switzerland. [5]Translational Sensory and Circadian Neuroscience, Max Planck Institute for Biological Cybernetics, Tübingen, Germany. [6]TUM Department Health and Sport Sciences, TUM School of Medicine and Health, Technical University of Munich, Munich, Germany. [7]TUM Institute for Advanced Study (TUM-IAS), Technical University of Munich, Garching, Germany. ✉e-mail: christine.blume@unibas.ch; manuel.spitschan@tum.de

to acute neuroendocrine responses in constant light[12] at ~170 lux, and there is evidence that cones are not necessary for circadian and neuroendocrine responses to light as some visually blind individuals suppress melatonin in response to light[13,14]. Thus, under continuous lighting conditions, circadian and neuroendocrine responses to light are largely driven by melanopsin[4,6].

## Prior evidence for S-cone opponent circuitry

In parallel to the physiological opponency of cone signals established using electrophysiological recordings in the primate retina[15], there is ample evidence that human image-forming colour vision is also organized according to colour-opponent channels. Signals from the three cone classes with overlapping but distinct spectral sensitivities (Fig. 1b) are recombined into three channels[16,17]: an additive combination of L and M cones (L + M; luminance), an opponent, subtractive combination of L and M cones (L − M; red−green), and an opponent combination pitting S cones against luminance (S − (L + M); blue−yellow; Fig. 1c). These post-receptoral channels form the basic dimensions of human colour vision, also called the cardinal directions of colour space[18]. It has previously been suggested, but not directly tested, that this circuit may participate in the suppression of nocturnal melatonin secretion by light[19].

## Dawn and dusk changes encoded by S-cone opponent circuitry

As the transition between day and night, twilight represents a key change in the light environment with changes in light intensity being largest during dusk and dawn[20]. Importantly, however, the spectral composition of environmental illumination also changes, with a boost in short-wavelength illumination relative to long-wavelength light[20,21]. This signal has long been hypothesized to be informative for the timing of activity and circadian photoentrainment, although this precise relationship has not yet been established[21–23]. In general, a colour-opponent system pitting short-wavelength signals against long-wavelength signals is well suited to pick up the spectral changes at dawn and dusk[22], but whether there is a dedicated colour-opponent input into the human circadian clock has presently not been demonstrated.

## Evidence for a +S − L opponent channel in mice

Recently, it was reported that, in mice, stimuli defined along the post-receptoral axis pitting murine S cones against murine L cones differ in their circadian effect, with 'yellow' and bright light (with high L-cone activation and low S-cone activation, activating a +S − L opponent channel) induced a stronger circadian phase-shift, as assessed by behavioural responses, than 'blue' and dim light (with lower L-cone activation and high S-cone activation, activating an −S + L opponent channel)[24]. This is somewhat counter-intuitive in the light of prior research as many studies in humans have shown that short-wavelength (enriched) light exerts stronger effects on the circadian system[1,19,25,26]. Although it is unknown whether this effect translates to humans, primate ipRGCs are characterized by an +S − (L + M) opponent organization as well[8,9], thereby providing a homologous substrate for the effects seen in mice. Based on these converging lines of evidence (for a summary of the extant literature, see Supplementary Table 1), we therefore hypothesized that a similar effect can be found in the circadian system of humans.

In this Registered Report, we examined the circadian phase-shifting effects of evening light exposure incorporating calibrated changes along the +S − (L + M), blue−yellow dimension of human vision and its effects on sleep in the ensuing night. Specifically, we examined circadian phase shifts in the melatonin rhythm in response to 1-h light exposure to three different scenarios in a 32.5-h, within-subjects in-laboratory protocol under controlled lighting and temperature conditions as well as its effects on acute melatonin suppression, subjective and objective sleepiness, visual comfort, psychomotor vigilance, sleep onset and slow-wave activity (SWA) during sleep (Fig. 2).

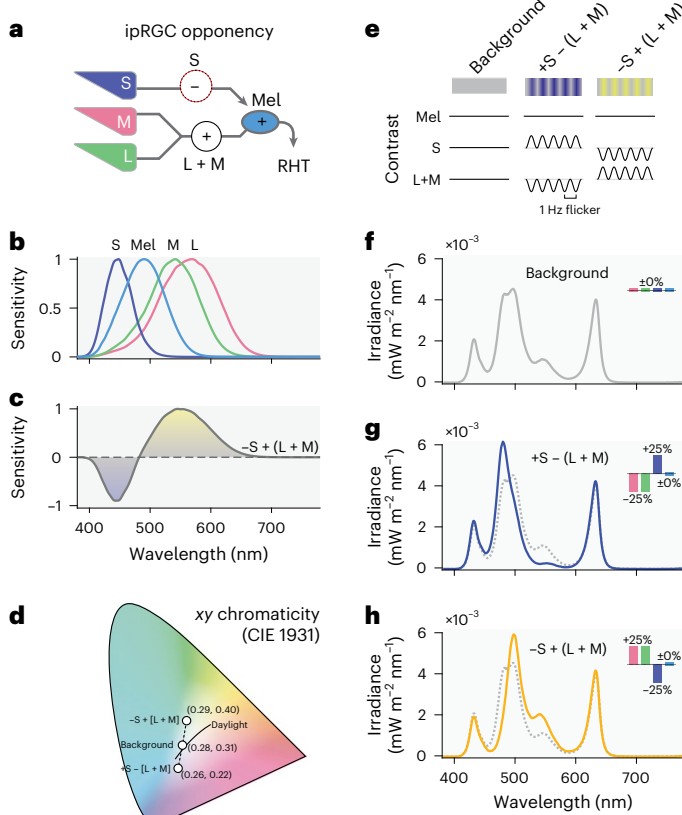

**Fig. 1 | S-cone-opponent influences into ipRGCs and selective stimuli to target these pathways. a**, The cone inputs to the melanopsin-containing ipRGCs in the human retina combine light information in an opponent, subtractive fashion, pitting signals from the S cones against signals a joint L- and M-cone signal encoding luminance (schematic diagram, simplifying the underlying anatomy; Mel = melanopsin-containing ipRGC, RHT = retinohypothalamic tract). **b**, The spectral sensitivities of the L, M and S cones have distinct peaks ($\lambda_{max}$) but are overlapping. **c**, The joint spectral sensitivity of the opponent −S + (L + M) channel, showing distinct wavelength regions yielding positive versus negative activations. **d**, Chromaticity diagram (CIE 1931 $xy$ chromaticity) showing the chromaticity coordinates of the background condition (corresponds to colour temperature of 6,500 K daylight (D65)), the two modulation spectra and the daylight locus (daylight spectra between 4,000 K and 25,000 K). **e**, Overview of stimulus conditions and its contrast properties. Stimuli are unipolar sinusoidally flickering excursions from the background in the direction of the +S − (L + M) and −S + (L + M) poles of the blue−yellow channel. **f**, Spectral irradiance distribution of the constant background, keeping excitation constant for L, M and S cones and melanopsin. Inset: contrast for the L, M and S cones, and melanopsin for the modulation spectrum against the background spectrum. **g**, Spectral irradiance distribution of the +S − (L + M) stimulus, biasing S cones over luminance (blue solid line) against the background (dashed grey line). Inset: contrast for the L, M and S cones, and melanopsin for the modulation spectrum against the background spectrum. **h**, Spectral irradiance distribution of the −S + (L + M) stimulus, biasing S cones over luminance (blue solid line) against the background (dashed grey line). Inset: contrast for the L, M and S cones, and melanopsin for the modulation spectrum against the background spectrum.

Stimuli were presented in wide-field, Newtonian (no artificial pupil) viewing without pharmacological dilation as sinusoidally flickering modulations (1 Hz, 30 s on, 30 s off to avoid adaptation) either favouring S cones over luminance (+S − (L + M); blue-dim) or luminance over S cones (−S + (L + M); yellow-bright) with no change in melanopsin activation around a background mimicking daylight (D65, 6,500 K; planned 100 lux (measured: 93.5 lux)). Additionally, we also examined the circadian response to this constant background.

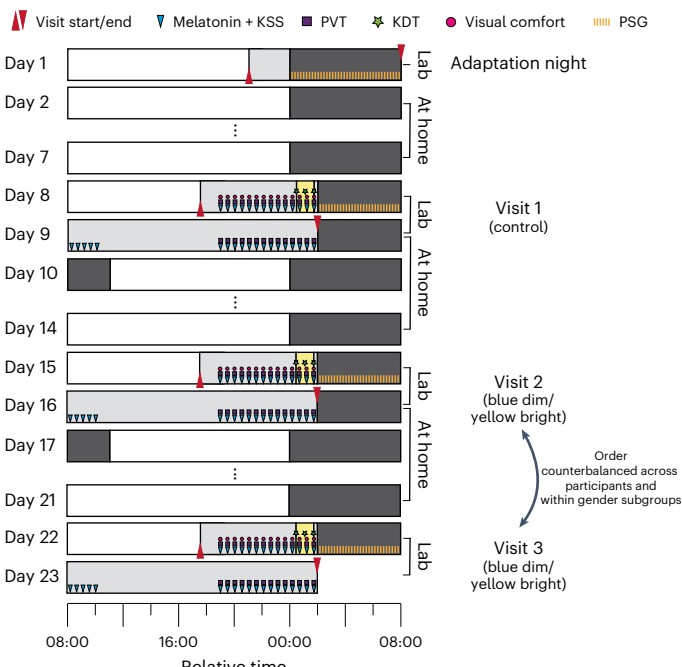

**Fig. 2 | Experimental protocol for within-subjects assessment of light-exposure conditions.** Participants engaged in a circadian stabilization protocol for a total of 23 days, during which they visited the laboratory for a total of four times: One adaptation night (day 1) and three 32.5-h protocol visits to assess the effect of the light exposure (1 h each, starting 30 min after HBT, as indicated by the yellow boxes) conditions on the human circadian clock. The first 32.5-h visit (protocol 1 on days 8 and 9) was always the constant background ('control') light, while the subsequent two visits were balanced between participants ($n = 16$ in total, 8 women and 8 men). For the experimental visits, volunteers arrived at the lab 6.5 h before their HBT. Melatonin sampling and assessments of subjective sleepiness (with the KSS), psychomotor vigilance (with the PVT) as well as assessments of visual comfort in 30-min intervals started 5 h before HBT. Before, during and after light exposure, we also assessed objective sleepiness with the KDT. Subsequently, participants went to bed for a 6-h sleep opportunity. Following wake-up, another five melatonin samples as well as KSS assessments were obtained. During the second evening, PVT and KSS measurements were performed as on evening 1. During all nights, PSG was recorded. The figure shows the relative time for a participant whose habitual bed time is at midnight. Grey boxes indicate dim light (that is, mostly <8 lux).

## Primary, confirmatory hypothesis

The primary outcome variable of this study was the circadian phase shift in salivary melatonin onset in the three conditions, constant light (always condition 1), blue dim flickering and yellow bright flickering, which were administered to each participant in a within-subjects design (for a description of the outcome variables, see Supplementary Tables 2 and 3). We specifically hypothesized that yellow-bright flickering would induce a larger circadian phase shift than blue-dim changes and that flickering generally induces larger shifts than constant light (yellow-bright > blue-dim > constant background; confirmatory test $C_1$; Supplementary Table 2).

Our hypothesis that flickering light would elicit a stronger effect was motivated by the psychophysical and physiological evidence[27-33] that cones and post-receptoral mechanisms adapt under constant conditions, as well as a recent study showing that flashing light causes a stronger phase shift than continuous light under certain conditions[34]. The hypothesis that yellow-bright flickering stimuli have a stronger effect than blue-dim stimuli was based on the previously discussed animal study by Mouland and colleagues[24]. We subjected the circadian phase shift to a Bayes factor test in a within-subjects analysis of variance (ANOVA) design and used the sequential Bayes factor[35] estimation

procedure to assess the evidence for the full model (incorporating the light condition type) over a null model (not incorporating the light condition type). We had planned to continuously recruit participants until the stopping criterion, or resource limit, would be reached. The stopping criterion was a Bayes factor (that is, $BF_{10}$ or $BF_{01}$) of ≥10, which corresponds to the full model (that is, light condition making a difference) being 10× more likely than the null model or vice versa. Our financially driven resource limit was $n = 16$ participants. Assuming a large effect size (ES; Cohen's $d = 0.8$) and that H1 better predicts the data than H0, simulations[36] revealed that 62% of the simulations showed evidence for H1 with 38% being inconclusive (medium ES: 22.2% versus 77.7%; small ES: 0.5% versus 96.4% and 3.1% showing evidence for H0). However, we argue that, if the $S - (L + M)$ opponent system indeed exerts a very strong effect on the circadian system, then this should be visible on a single-subject level already. In fact, previous research suggests that the assumptions about ESs (that is, $d = 0.8$) for the circadian phase shifts used in the simulations outlined above may be rather conservative[37]. Thus, if the results remain inconclusive even after 16 participants, we argue that, if there is an effect we have missed, such an effect is not physiologically meaningful in the average healthy individual. For the field, this is of great importance, as it would directly inform that any influence of $S - (L + M)$ stimulation on the circadian system is very small, or subject to very strong interindividual differences. In those cases, the message is that melanopsin is the main driver of circadian photoentrainment.

## Secondary, ring-fenced hypotheses

In addition to our primary outcome, we also examined a set of secondary endpoints (for a description of the outcome variables and associated hypotheses, see Supplementary Tables 2 and 3). The time series of melatonin concentrations in the evening of the first night was subjected to a repeated-measures ANOVA, and the evidence for a difference in the acute melatonin-suppressive effect of light was examined using Bayes factors (S1). We hypothesized that the yellow-bright stimuli would have a larger melatonin-attenuating effect than the blue-dim stimuli and that both would have a larger effect than constant light. Psychometrically, we collected sleepiness ratings using the Karolinska Sleepiness Scale (KSS)[29] (S2) and visual comfort ratings using a custom nine-item rating scale (S4), which we likewise planned to subject to a Bayesian repeated-measures ANOVA to examine differences. Objective sleepiness was assessed by ratio of absolute theta (4–7 Hz) and alpha (8–12 Hz) power at parietal and occipital electrodes during resting-state electroencephalography (EEG) with both eyes open[38] (S3). We hypothesized that sleepiness (ratings) would be lower and visual discomfort ratings would be higher in the yellow-bright stimuli than in the blue-dim stimuli. Constant light was expected to be associated with highest sleepiness (ratings) and highest visual comfort ratings. We also examined changes in performance on a reaction time (RT) test, a modified auditory psychomotor vigilance test (PVT)[25,26,39-41], to test for differences in vigilant attention between the three lighting conditions using a Bayesian repeated-measures ANOVA. We hypothesized that yellow-bright stimuli would produce faster RTs as measured with the PVT than blue-dim stimuli and that constant light would be associated with slowest RTs (median, 10% fastest, 10% slowest; S5, S6 and S7). Finally, we examined properties of sleep using polysomnography (PSG) in the night following light exposure. We specifically examined sleep onset latency (SLAT) to constant 10 min of sleep[37,42] (S8), which we hypothesized to be longer in the yellow-bright than in the blue-dim condition and shortest in the constant light condition. We also studied SWA (delta power density between 0.5 Hz and 4.5 Hz, S9) during the first sleep cycle[25], which we hypothesized to be decreased in the yellow-bright condition compared with the blue-dim condition and highest in the constant light condition.

## Results

To characterize spectral shifts of the display with different driving input, we measured the spectrum of each of the primaries using a Jeti

**Table 1 | Stimulus characteristics**

| | α-opic irradiances (mW m⁻²) | | | | CIE 1931 *xyY* | | |
| --- | --- | --- | --- | --- | --- | --- | --- |
| | **L cones** | **M cones** | **S cones** | **Melanopsin** | ***x*** | ***y*** | **Illuminance (lux)** |
| Background | 159.22 | 160.83 | 84.90 | 212.91 | 0.25 | 0.30 | 93.53 |
| +S−(L+M) | 123.67 | 124.21 | 110.63 | 219.60 | 0.23 | 0.20 | 67.01 |
| Contrast | −22.33% | −22.77% | 30.31% | 3.15% | | | |
| −S+(L+M) | 200.71 | 204.06 | 63.21 | 216.83 | 0.26 | 0.40 | 123.49 |
| Contrast | 26.06% | 26.88% | −25.54% | 1.84% | | | |
| Daylight (D65) | 162.89 | 145.57 | 81.71 | 132.60 | 0.31 | 0.33 | 100 |

Overview of the irradiance-derived α-opic responses (in mW m⁻²) for the three experimental screen light conditions with ambient dim light. Photopic illuminance was 93.5 lux (background), 123.5 lux (yellow-bright) and 67.1 lux (blue-dim). Radiance-derived chromaticity values (CIE 1931 *xy* standard observer for a 2° field) were *x*=0.25 and *y*=0.3 for the background, *x*=0.26 and *y*=0.4 for the yellow bright, and *x*=0.23 and *y*=0.2 for the blue-dim condition. Measures were taken at a distance of 68 cm from the screen at a height of 118 cm, that is, from the observer's point of view. Values were calculated using the luox app[97,98]. For a reference, we also report values for daylight (D65) at 93.5 lux.

spectraval 1501 (JETI Technische Instrumente GmbH). For an overview of the irradiance and radiance-derived α-opic responses for the three experimental screen light conditions and the corresponding irradiance and radiance-derived equivalent daylight (D65) illuminances, melanopic equivalent daylight illuminance (EDI), as well as luminances, please see Tables 1–4. For an overview of the irradiance-derived α-opic responses, equivalent daylight (D65) illuminances (in lux), and melanopic EDI of the light during the scheduled day, please see Supplementary Tables 4 and 5.

In total, 47 individuals completed the online screening, and 18 participants were invited to participate in the study. Two volunteers decided to not continue with the study after the adaptation night for personal reasons. Thus, 16 healthy, young male and female participants (mean 25.5 ± 2.7 years; 8 men and 8 women) were recruited. Data acquisition took place continuously between March and December 2022 at the Centre for Chronobiology at the University of Basel with a break of 4 weeks in August 2022. For more details on the distribution of participants across the acquisition period including subjectively reported light history on the day of the experimental visit, please see Supplementary Information and the laboratory log. The order of the light exposure conditions for each participant was determined by the participant number, that is, uneven and even participant numbers were associated with one or the other order. The participant number was assigned by C.B., and numbers were assigned in the order of enrolment. The final decision to enrol a participant was the responsibility of the first author (C.B.), in some cases after consultation with the study clinician (H.C.S.) and the senior author (M.S.). Participants always entered the lab on the same day of the week.

For all analyses, we report BF₁₀, that is, the likelihood of the data under H1 compared with H0. For the interpretation of the BFs, please see Supplementary Table 6. All analyses were based on data from *n* = 16 participants. Where data from single conditions were missing, this has been detailed below.

### Primary outcome

**DLMO; C1.** To determine the dim-light melatonin onset (DLMO), we used all available evening data points and applied the 'Hockey-stick' algorithm[85]. We chose a default area of interest upper border or threshold of 5 and selected 'Hockeystick Time'. In rare cases, the threshold had to be adapted for some individuals. C.B., C.C. and M.S. independently performed the DLMO analyses with CC and MS being blinded to the light exposure conditions. Any divergences were resolved in a final discussion with Dr Mirjam Münch, a colleague at the Centre for Chronobiology, who is very experienced with applying the 'Hockeystick' algorithm. In two cases (1× background, 1× blue-dim condition), it was not possible to determine a DLMO on at least one evening as no clear increase in melatonin concentrations could be identified. The inspection of the data showed that the assumptions

**Table 2 | Overview of the irradiance-derived equivalent daylight (D65) illuminances (EDI; in lux) for the three experimental screen light conditions with ambient dim light**

| | Equivalent daylight illuminance (lux) | | | |
| --- | --- | --- | --- | --- |
| | **L cones** | **M cones** | **S cones** | **Melanopsin** |
| Background | 97.75 | 110.74 | 103.88 | 160.54 |
| +S−(L+M) | 75.92 | 85.32 | 135.36 | 165.59 |
| Contrast | −22.33% | −22.95% | 30.30% | 3.05% |
| −S+(L+M) | 123.22 | 140.17 | 77.34 | 163.50 |
| Contrast | 26.06% | 26.58% | −25.55% | 1.81% |

Measures were taken at a distance of 68 cm from the screen at a height of 118 cm, that is, from the observer's point of view. Values were calculated using the luox app[97,98].

for mixed linear models were met (for more details, see Supplementary Methods).

Contrasting our primary hypothesis, analyses yielded moderate and thus inconclusive evidence against the hypothesis that there was a condition difference (BF₁₀ = 0.3). The data were approximately 3.4 times more likely to occur under H0 than under H1. The mean phase shift was 52.0 min in the background condition (range −42.3 to 164.4 min), 41.94 min in the blue-dim condition (range −22.8 to 104.1 min) and 33.8 min in the yellow-bright condition (range −29.7 to 101.7 min). Supplementary Table 7 provides an overview of the condition mean (intercept) and deviations from the intercept sampled from the posterior distribution. For a graphical illustration of the results, please see Fig. 3a. For an illustration of the melatonin profiles on evenings 1 and 2 in each condition, see Supplementary Fig. 1.

### Secondary outcomes

**Melatonin concentrations (S1).** The inspection of the data showed that rank transformation of the data resulted in an increased compatibility with the assumptions for mixed linear models. Thus, we additionally report results for rank-transformed data (for details, see Supplementary Methods). Analyses of melatonin values yielded strong evidence against a condition difference during the light exposure (BF₁₀ = 0.09) suggesting that the data were approximately 11 times more likely to occur under H0. This result was largely confirmed using analyses based on rank-transformed data, which yielded moderate evidence against H1 (BF₁₀ rank-based = 0.14). For an overview of the condition mean (intercept) and deviations from the intercept sampled from the posterior distribution, please see Supplementary Table 8. There was moderate (inconclusive) evidence against a condition × time interaction (BF₁₀ = 0.13; BF₁₀ rank-based = 0.13). Please see Fig. 3b for an illustration of the results and Supplementary Fig. 1 for an illustration of the melatonin concentrations on evenings 1 and 2.

**Table 3 | Overview of the radiance-derived α-opic responses (in mW m⁻² sr⁻¹) for the three experimental screen light conditions with ambient dim light**

| | α-opic radiances (mW m⁻² sr⁻¹) | | | | CIE 1931 xyY | | |
|---|---|---|---|---|---|---|---|
| | L cones | M cones | S cones | Melanopsin | x | y | Luminance (cd m⁻²) |
| Background | 508.24 | 543.67 | 304.10 | 755.63 | 0.25 | 0.30 | 298.92 |
| +S−(L+M) | 377.36 | 410.45 | 394.49 | 775.19 | 0.23 | 0.20 | 202.32 |
| Contrast | −25.75% | −24.50% | 29.72% | 2.52% | | | |
| −S+(L+M) | 653.75 | 696.25 | 224.66 | 767.68 | 0.26 | 0.40 | 404.37 |
| Contrast | 28.63% | 28.06% | −26.12% | 1.59% | | | |

Photopic luminance was 298.92 cd m⁻² (background), 404.37 cd m⁻² (yellow-bright) and 202.32 cd m⁻² (blue-dim). Radiance-derived chromaticity values (CIE 1931 xy standard observer for a 2° field) were x=0.25 and y=0.3 for the background, x=0.26 and y=0.4 for the yellow-bright, and x=0.23 and y=0.2 for the blue-dim condition. Measures were taken at a distance of 68 cm from the screen at a height of 118 cm, that is, from the observer's point of view. Values were calculated using the luox app[97,98]. cd = candela, SI-unit for luminous intensity.

**Table 4 | Overview of the radiance-derived equivalent daylight (D65) luminances (EDL; in cd m⁻²) for the three experimental screen light conditions with ambient dim light**

| | Equivalent daylight luminance (cd m⁻²) | | | |
|---|---|---|---|---|
| | L cones | M cones | S cone | Melanopsin |
| Background | 312.01 | 373.45 | 372.09 | 569.77 |
| +S−(L+M) | 231.73 | 281.94 | 482.68 | 584.51 |
| Contrast | −25.73% | −24.50 | 29.72% | |
| −S+(L+M) | 401.34 | 478.25 | 274.88 | 578.85 |
| Contrast | 28.63% | 28.06% | −26.13% | |

Measures were taken at a distance of 68 cm from the screen at a height of 118 cm, that is, from the observer's point of view. Values were calculated using the luox app[97,98]. cd = candela, SI-unit for luminous intensity.

**Subjective sleepiness (S2).** Also here, the inspection of the data showed that rank transformation resulted in an increased compatibility with mixed linear model assumptions (for details, see Supplementary Methods). Thus, we additionally report results for rank-transformed data. Analyses yielded anecdotal and thus inconclusive evidence against a condition difference regarding the subjectively reported sleepiness on the KSS during the light exposure. Under the H0, the data were approximately 1.6 times more likely than under the H1 ($BF_{10} = 0.62$; $BF_{10\,rank\text{-}based} = 0.23$). For the condition mean (intercept) and deviations from the intercept sampled from the posterior distribution, please see Supplementary Table 9. Furthermore, there was moderate (inconclusive) evidence against a time × condition interaction ($BF_{10} = 0.27$; $BF_{10\,rank\text{-}based} = 0.14$). For a graphical illustration, see Fig. 3d, and for an illustration of the KSS values on evenings 1 and 2 in each light condition, please see Supplementary Fig. 6 (note this was not part of the analysis plan, wherefore we refrain from a statistical analysis).

**Objective sleepiness (S3).** We assessed objective sleepiness, that is, the alpha (8–12 Hz) to theta (4–7 Hz) ratio during a 3-min Karolinska Drowsiness Test (KDT) with eyes open at the beginning, after 30 min, and at the end of the 1-h light exposure at parieto-occipital electrodes. The inspection of the data indicated that rank transformation resulted in an increased compatibility with the model assumptions. Thus, we additionally report results for rank-transformed data (for details, see Supplementary Information). There was moderate (inconclusive) evidence against a condition effect with the data being approximately 3.5 times more likely under H0 than under H1 ($BF_{10} = 0.28$; $BF_{10\,rank\text{-}based} = 0.09$). For an overview of the condition mean (intercept) and deviations from the intercept sampled from the posterior distribution, please see Supplementary Table 10. Analyses additionally yielded strong evidence against a time × condition interaction ($BF_{10} = 0.06$; $BF_{10\,rank\text{-}based} = 0.08$).

**Visual comfort (S4).** Visual comfort was calculated as the average rating from the responses to the questions about how pleasant the lighting was generally, how participants perceived the level of brightness, how glaring the light source was, and how pleasant participants rated the colour temperature (that is, cold vs. warm). The inspection of the data showed that the assumptions for mixed linear models were met (for more details, see Supplementary Methods). There was moderate (inconclusive) evidence against a difference between the conditions regarding visual comfort experienced during the light exposure. The data were approximately six times more likely under the H0 than under the H1 ($BF_{10} = 0.16$). Supplementary Table 11 provides an overview of the condition mean (intercept) and deviations from the intercept sampled from the posterior distribution. For the condition × time interaction, there was moderate evidence in favour of H1 ($BF_{10} = 7.68$). For a graphical illustration, see Supplementary Fig. 2.

**PVT: median RT (S5).** The inspection of the data suggested that the assumptions for mixed linear models were largely met, and rank transformation of the data did not improve compatibility (for more details, see Supplementary Methods). Analyses revealed extreme evidence in favour of a condition difference during the light exposure with the data being approximately 153 times more likely under the H1 than the H0 ($BF_{10} = 153.33$). More precisely, there was conclusive evidence in favour of faster median RTs in the background ($\text{mean}_{background}$ 384.4 ± 35.5 ms) than the yellow bright ($\text{mean}_{yellow\text{-}bright}$ 395.5 ± 32.6 ms; $BF_{10} = 29.06$) or the blue dim condition ($\text{mean}_{blue\text{-}dim}$ 399.8 ± 39.1 ms; $BF_{10} = 260.1$). There was only anecdotal (inconclusive) evidence against a difference between the yellow-bright and the blue-dim conditions ($BF_{10} = 0.39$). Supplementary Table 12 provides an overview of the condition mean (intercept) and deviations from the intercept sampled from the posterior distribution. There was anecdotal evidence against a condition × time interaction with the data being approximately two times more likely under the H0 compared with H1 ($BF_{10} = 0.47$). For a graphical illustration, see Fig. 3c and Supplementary Figs. 3 and 4 additionally provides an illustration of the PVT results on evenings 1 and 2 (note this was not included in the analysis plan, wherefore we refrain from statistical analyses).

**PVT: fastest 10% RTs (S6).** The inspection of the data suggested that the assumptions for mixed linear models were met and transformation of the data did not increase compatibility with the assumptions (for more details, see Supplementary Methods). For the 10% fastest RTs during the light exposure, there was only anecdotal and thus inconclusive evidence in favour of a condition difference with the data being approximately 1.8 times more likely under H1 compared with H0 ($BF_{10} = 1.77$). Supplementary Table 13 provides an overview of the condition mean (intercept) and deviations from the intercept sampled from the posterior distribution. Regarding the condition × time interaction, there was anecdotal evidence against H1 ($BF_{10} = 0.42$). Supplementary Fig. 3 provides an illustration of the results. Supplementary Fig. 4

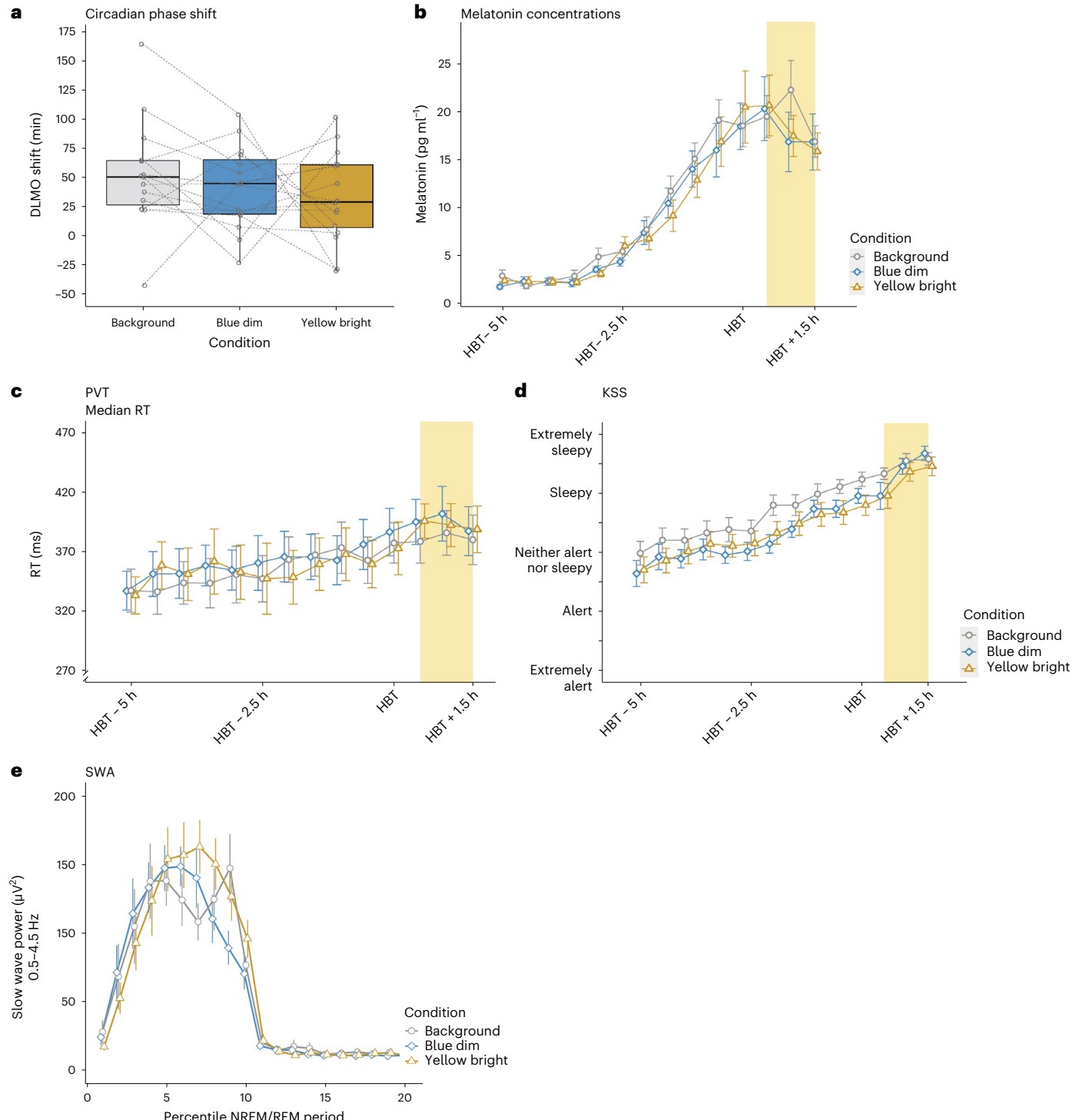

**Fig. 3 | No conclusive evidence for an effect of calibrated changes in light colour along the blue–yellow axis at constant melanopic excitation. a**, Circadian phase shift as assessed by a shift in DLMO from evening 1 to evening 2. Analyses yielded inconclusive evidence against the hypothesis of a condition difference (for details, see main text). The lower and upper hinges of the boxplots correspond to the 25% and 75% quartiles; the thick black line indicates the median. Whiskers extend to the lowest/largest value at most 1.5× the interquartile range from the hinges. Grey circles represent individual values of participants, and the dotted lines connect data points contributed by each participant. Analyses were based on data from 16 participants who underwent 3 experimental conditions. However, in two cases (1× background, 1× blue-dim), it was not possible to determine the DLMO on both evenings. **b**, Time course of melatonin concentrations during the first evening in the laboratory. We show the mean with error bars representing the standard error. Analyses were based on data from 16 participants who underwent 3 experimental

conditions. **c**, Time course of the median RTs on the PVT during the first evening in the laboratory. The figure shows median RTs and 95% confidence intervals. Analyses were based on data from 16 participants who underwent 3 experimental conditions. **d**, Time course of subjective sleepiness ratings on the KSS during the first evening in the laboratory. The figure shows mean RTs and error bars indicate standard errors. Analyses were based on data from 16 participants who underwent 3 experimental conditions. **e**, SWA as an indicator of homeostatic sleep pressure across the first sleep cycle. The time course relates to the percentiles of the NREM period (numbers 1–10 on the x axis) and the REM period (numbers 11–20 on the x axis) of the first sleep cycle after sleep onset. The data points represent the mean slow wave power between 0.5 Hz and 4.5 Hz with error bars indicating standard errors. Analyses were based on data from 16 participants who underwent 3 experimental conditions. However, due to data quality issues, we were unable to determine SWA in one case (yellow-bright condition).

additionally provides an illustration of the PVT results on evenings 1 and 2 (note this was not included in the analysis plan, wherefore we refrain from statistical analyses).

**PVT: slowest 10% RTs (S7).** Rank transformation of the data improved especially normality of the residuals and homoscedasticity. Thus, we additionally report analyses based on rank-transformed data. For the slowest 10% RTs during the light exposure, there was conclusive evidence in favour of H1, that is, a condition difference ($BF_{10} = 60.9$; $BF_{10 \, \text{rank-based}} = 119.8$). As for the median RTs, there was conclusive evidence in favour of RTs being faster in the background ($mean_{background}$ $481.6 \pm 67.8$ ms) compared with the yellow-bright ($mean_{\text{yellow-bright}}$ $511.0 \pm 81.6$ ms; $BF_{10} = 22.6$; $BF_{10 \, \text{rank-based}} = 40.86$) and the blue-dim conditions ($mean_{\text{blue dim}} 516.9 \pm 87.4$ ms; $BF_{10} = 46.15$; $BF_{10 \, \text{rank-based}} = 67.31$). However, there was moderate (inconclusive) evidence against a condition difference between the blue-dim and the yellow-bright conditions ($BF_{10} = 0.26$; $BF_{10 \, \text{rank-based}} = 0.23$). Supplementary Table 14 provides an overview of the condition mean (intercept) and deviations from the intercept sampled from the posterior distribution. There was moderate to strong evidence against a condition × time interaction ($BF_{10} = 0.09$; $BF_{10 \, \text{rank-based}} = 0.1$). For a graphical illustration, see Supplementary Fig. 3. Supplementary Figure 4 additionally provides an illustration of the PVT results on evenings 1 and 2 (note this was not included in the analysis plan, wherefore we refrain from statistical analyses).

**EEG-derived SLAT (S8).** The inspection of the data indicated that the assumptions for mixed linear models were met, and rank transformation did not increase compatibility with the assumptions (for more details, see Supplementary Methods). Analyses yielded inconclusive evidence against a condition difference regarding the onset latency to 10 min of continuous sleep. The data were approximately theee times more likely given the H0 than the H1 ($BF_{10} = 0.31$). The mean latency to 10 min of continuous sleep was $17.0 \pm 42.2$ min in the background, $9.4 \pm 13.2$ min in the yellow-bright, and $7.5 \pm 4.3$ min in the blue-dim condition. An overview of the condition means (intercept) and standard deviations sampled from the posterior distribution is provided in Supplementary Table 15. Supplementary Fig. 5 provides an illustration of the results. For a comprehensive overview of the sleep data for each light condition and visit, see Supplementary Table 16.

**EEG-derived SWA (S9).** Data inspection suggested that the assumptions for mixed linear models were met. Rank-transforming the data did partially result in an increased compatibility with the assumptions, wherefore we report results from analyses based on rank-transformed data (for more details, see Supplementary Methods). Note that during two nights one participant had a very short rapid eye movement (REM) sleep latency (that is, <30 min), which would have resulted in a very short non-REM (NREM) part. We thus decided to combine the NREM parts of the detected first and second sleep cycles into one. Please also note that in one participant we were unable to analyse the data in one condition (yellow-bright) as it was too noisy. Analyses of SWA (0.5–4.5 Hz) at frontal electrodes F3, Fz and F4 during the NREM part of the first sleep cycle yielded conclusive evidence against a condition difference ($BF_{10} = 0.03$; $BF_{10 \, \text{rank-based}} = 0.03$). Thus, the data were approximately 33 times more likely under the H0 than under H1. Evidence for a time × condition difference remained inconclusive (anecdotal evidence against H0; $BF_{10} = 0.42$; $BF_{10 \, \text{rank-based}} = 0.43$). For a graphical illustration, see Fig. 3e. Supplementary Table 17 provides an overview of the condition mean (intercept) and deviations from the intercept sampled from the posterior distribution.

**Outcome-neutral measurements**
**Melatonin concentrations (ON1).** The inspection of the data showed that rank transformation of the data slightly increased compatibility with the assumptions for mixed linear models. We therefore

additionally report results for rank-transformed data (for details, see Supplementary Information). There was strong evidence against a condition difference in melatonin concentrations before the beginning of the light exposure ($BF_{10} = 0.03$; $BF_{10 \, \text{rank-based}} = 0.03$) suggesting that the data were approximately 33 times more likely under the H0 than under the H1. Supplementary Table 18 provides an overview of the condition mean (intercept) and standard deviations sampled from the posterior distribution. For the condition × time interaction, there was extreme evidence against H1 ($BF_{10} = 0.003$; $BF_{10 \, \text{rank-based}} = 0.003$). For a graphical illustration, see Fig. 3b.

**PVT: median RT (ON2).** The inspection of the data suggested that the assumptions for mixed linear models were largely met, and rank transformation of the data did not improve compatibility (for details, see Supplementary Methods). Before beginning of the light exposure, analyses yielded strong evidence in favour of a condition difference ($BF_{10} = 17.42$). Follow-up analyses indicated evidence in favour of faster RTs in the background ($mean_{background} 359.1 \pm 36.7$ ms) compared with the yellow-bright ($mean_{\text{yellow-bright}} 365.2 \pm 42.5$ ms) or the blue dim condition ($mean_{\text{blue-dim}} 365.8 \pm 37.9$ ms; $BF_{10} = 19.8$). There was only moderate inconclusive evidence in favour of a difference between the yellow-bright and the blue-dim conditions ($BF_{10} = 5.49$). For an overview of the condition mean (intercept) and standard deviations sampled from the posterior distribution please see Supplementary Table 19. Regarding a condition × time interaction, there was extreme evidence against H1 with the data being approximately 222 times more likely under the H0 compared with H1 ($BF_{10} = 0.004$). Figure 3c provides a graphical illustration of the results.

**Subjective sleepiness (ON3).** The inspection of the data showed that the assumptions for mixed linear models were met (for details, see Supplementary Information). For the subjective sleepiness before the beginning of the light exposure, there was extreme evidence in favour of H1, that is, a condition difference with the data being >100 times more likely under the H1 compared with H0 ($BF_{10} = 397,569,714$). More specifically, participants felt more tired in the background ($mean_{\text{KSS background}} 6.19 \pm 1.6$) compared with the yellow-bright condition ($mean_{\text{KSS yellow-bright}} 5.56 \pm 1.7$; $BF_{10} = 548,934$; extreme evidence in favour of H1) and to the blue-dim condition ($mean_{\text{KSS blue-dim}} 5.47 \pm 1.5$; $BF_{10} = 321,238,205$; extreme evidence in favour of H1). There was moderate (inconclusive) evidence against a condition difference between the yellow-bright and the blue-dim conditions ($BF_{10} = 0.16$; moderate evidence against H1). Supplementary Table 20 provides an overview of the condition means (intercept) for the subjective sleepiness scores and standard deviations sampled from the posterior distribution. Analyses further yielded very strong evidence against a condition × time interaction with the data being approximately 86 times more likely under the H0 than under H1 ($BF_{10} = 0.01$). For a graphical illustration, see Fig. 3d.

**Brightness (ON4).** The inspection of the data showed that the assumptions for mixed linear models were rather not met, particularly the residuals were not normally distributed (for details, see Supplementary Information). However, a rank transformation did not solve this issue, wherefore we refrain from reporting analyses based on rank-transformed data. There was extreme evidence in favour of a condition difference between the three light exposure conditions before the light exposure ($BF_{10} = 110.93$) with the data being approximately 111 times more likely under the H1 than the H0. The mean perceived brightness (± standard deviation) in the background condition (range 1–5) was $3.3 \pm 0.56$, in the blue-dim condition $3.4 \pm 0.53$, and in the yellow-bright condition $3.5 \pm 0.55$. More specifically, there was extreme evidence in favour of a condition difference between the background and the yellow-bright conditions ($BF_{10} = 752.3$), moderate (inconclusive) evidence against a difference between background and the blue-dim conditions ($BF_{10} = 0.31$) and moderate (inconclusive) evidence in favour

of a difference between the blue-dim and yellow-bright conditions ($BF_{10} = 7.72$). For an overview of the condition mean (intercept) and standard deviations sampled from the posterior distribution, please see Supplementary Table 21. For the condition × time interaction, there was anecdotal (inconclusive) evidence against an effect ($BF_{10}$ 0.36).

## Discussion

In this Registered Report, we found no conclusive evidence for an effect of calibrated silent-substitution changes in light colour along the blue–yellow axis on the human circadian clock or sleep. In a targeted test of our primary hypothesis, there was no conclusive evidence for differential phase-delaying effects of a 1-h nocturnal light exposure (starting 30 min after habitual bedtime, HBT) to constant background/control light, blue-dim and yellow-bright flickering stimuli using moderate light levels typical for room illumination and constant melanopic excitation across light conditions. Additionally, there was conclusive evidence against differences in melatonin suppression. Thus, we conclude that, even if there is an effect we have missed, the contribution of a post-receptoral channel, where S-cone signals are pitted against a joint $L + M$ signal (that is, luminance; $S - (L + M)$), is probably not physiologically relevant to the circadian timing system in healthy young humans at night under typical room illuminance levels. Rather, this underscores a probably primary role of melanopsin-containing ipRGCs in mediating these effects as has previously been demonstrated repeatedly, for example, refs. 43–46. In line with this, a recent meta-analysis[47] concluded that melanopic EDI was the best single predictor for melatonin suppression at light levels above 21 photopic lux. Furthermore, there was no conclusive evidence for differential effects of the calibrated changes in light colour on visual comfort, subjective and objective sleepiness, psychomotor vigilance or EEG-derived sleep latency, while there was conclusive evidence against differential effects regarding homeostatic sleep pressure as assessed by SWA during the first sleep cycle.

In more detail, the 1-h light exposure starting 30 min after HBT induced a phase delay in all three light exposure conditions suggesting that the light exposure generally had an effect on the circadian timing system (BFs from post hoc *t*-tests assessing difference from zero: $BF_{background} = 59.5$; $BF_{blue-dim} = 14.8$; $BF_{yellow-bright} = 86.8$). Given that the unmasked endogenous period length under dim light conditions has been suggested to be about 24.3 h (ref. 48), it seems unlikely that the observed phase delays can solely be attributed to the lack of morning light. However, evidence for the hypothesis that yellow-bright flickering light would induce a larger phase delay than blue-dim light as well as that flickering light would generally lead to larger shifts than the background condition, remained inconclusive. Our results contrast earlier findings in mice by Mouland and colleagues[24] who reported that a light stimulus biased towards S-opsin activation thus appearing blue resulted in weaker circadian responses than yellow stimuli with a bias towards L-cone activation when rod and melanopsin activation were held constant. Specifically, in their study, the period length as indicated by the length of the mice's rest–activity cycles was prolonged under yellow light, suggesting a constant stronger phase shift than in the blue condition. Apart from the difference in the studied species, one explanation for the deviating findings could be differences in the duration and timing of the light exposure. In Mouland et al.'s study, mice were exposed to the yellow or blue light at all times during the light phases[24], whereas we only used a 1-h light exposure at a specific circadian phase (that is, starting 30 min after habitual bed time). However, earlier findings by Gooley and colleagues[6] had suggested that cones substantially contribute to effects especially at the beginning of a light exposure and at rather low irradiance levels. Here, participants had been exposed to a narrow-bandwidth 555 nm green light or blue light at 460 nm (10–14 nm half-peak bandwidth) specifically targeting melanopsin for 6.5 h starting near the onset of nocturnal melatonin secretion (16 irradiance levels covering a broad range of photon densities randomized across participants; $2.52 \times 10^{11}$

to $1.53 \times 10^{14}$ photons $cm^{-2} s^{-1}$). Regarding the phase-resetting response, the authors concluded that the effects of the exposure to green relative to blue light were too large to solely be attributed to melanopsin excitation, suggesting an involvement of the cones. Likewise, both lights were equally effective in suppressing melatonin at the beginning of the exposure, with the importance of melanopsin increasing exponentially with the length of exposure. This notion has recently received further support from a publication resulting from the same study, where St. Hilaire and colleagues[49] investigated the phase-shifting and melatonin-suppressing responses in now six wavelength conditions (420 nm, 460 nm, 480 nm, 507 nm, 555 nm and 620 nm; photon densities ranging from $2.52 \times 10^{11}$ to $1.53 \times 10^{14}$ photons $cm^{-2} s^{-1}$)[49]. Here, the authors concluded that ipRGCs contributed 33%, S cones 51% and L + M cones 16% to the phase-shifting effects of the 6.5-h light exposure. Regarding melatonin suppression, during the first quarter of the 6.5-h light exposure (that is, 97.5 min), S cones and L + M cones allegedly substantially contributed to melatonin suppression (51% and 47% contributions of the respective spectral sensitivity curves to the overall sensitivity), while ipRGCs contributed only 2%. As in Gooley et al.[6], the contribution of ipRGCs and thus melanopsin substantially increased across time. In the light of these results, we argue that a 1-h exposure specifically targeting the cone-based post-receptoral mechanisms should have been sufficient to produce conclusive differential effects between the background condition and the blue-dim or yellow-bright conditions in the present study. Thus, our findings challenge the notion of a strong contribution of cone signals during shorter nocturnal light exposure put forward by Gooley et al.[6] and St. Hilaire et al.[49]. It is important to note that these studies deployed monochromatic lights of different wavelengths and intensities not well suited to stimulate a specific photoreceptor class and fit photoreceptor-based spectral sensitivities to the resulting empirical action spectra. In contrast, our stimulus design specifically targeted cone-based mechanisms, thereby cleanly isolating the potential contribution of the cones. Future and adequately powered studies using calibrated, photoreceptor-isolating light stimuli, varying irradiance levels, temporal properties and exposure duration will have to finally settle this question.

Furthermore, several investigations of the interaction between circadian phase and the magnitude and direction of the effects of light exposure have suggested increased sensitivity of the circadian timing system at the phase of exposure to bright[50,51] and blue-looking light[52]. Nevertheless, future studies will need to evaluate the effects of calibrated colour changes along the blue–yellow axis at different circadian phases, and different mean light levels as well as melanopic excitation levels, and to investigate whether there is a specific sensitivity to calibrated changes along the blue–yellow dimension, for instance specifically during the twilight hours.

Conclusive condition differences were limited to differences in psychomotor vigilance, that is, RTs on the psychomotor vigilance task (PVT) and subjective sleepiness as assessed with the KSS. Considering median RTs, participants were faster in the background condition than in the blue-dim or the yellow-bright conditions before as well as during the light exposure. During light exposure, this effect was also visible when looking at the 10% slowest responses. Especially because RT differences were not limited to the 1-h light exposure but present even before, this most probably seems an effect of increased motivation during the first experimental visit (always background condition). Interestingly and somewhat contradictorily, the faster RTs in the background condition coincided with higher subjective sleepiness ratings. This difference was however limited to the time before the light exposure. We speculate that this may have resulted from participants not knowing what to expect and how they would react to the prolonged wake time and the laboratory protocol, which may have been perceived as tiring.

Limitations of the study include the fact that the order of the conditions was not completely randomized, but the background condition was always the first one. This had logistical reasons as randomizing

three conditions would have resulted in six potential condition orders. Given the results from the Bayes factor design analysis[36], such an approach would possibly have required us to obtain data from 2 × 12 participants (that is, 2 × 6 men and 6 women), which would have exceeded the financial and resource limit of this project. Besides this, the fact that the light exposure took place between 30 min and 1 h 30 min after HBT may have resulted in homeostatic sleep pressure overriding more subtle effects of the different light exposure conditions. However, other studies investigating the effects of light, where participants likewise went to sleep well after HBT had still found differences between monochromatic light exposure conditions on for instance SWA at the beginning of the night[25,53], self-reported sleepiness[54] and psychomotor vigilance[54]. Furthermore, we did not objectively assess light history before the arrival at the lab, which is known to affect the response to a subsequent light stimulus[55,56]. However, evidence for differences between the conditions in self-reported time spent under the open sky before arriving at the lab ($BF_{10} = 0.33$; $BF_{10\,rank\text{-}based} = 0.16$), or self-reported lighting conditions outside on that day during the time they had spent under the open sky, remained inconclusive ($BF_{10} = 0.16$; 10-point Likert scale from 'very dull day' to 'bright summer's day;'). Additionally, participants' visits were scheduled within 3 weeks and took place on the same day of the week contributing to a maximal stability of ambient lighting conditions as well as individual schedules and thus light history. Furthermore, we did not control pupil size, which may have led to small changes in retinal irradiance. While it is well known that cone mechanisms contribute to the regulation of pupil size[10,57–60], it is implausible that small variations in retinal irradiance due to differences in pupil size would have counteracted any true photoreceptor-mediated effect. Spitschan et al.[10] examined the response of the pupil to flickering stimuli at different temporal frequencies, finding pupillary response amplitudes for 1 Hz stimulation of the L + M channel (50% contrast) around 8%. For a pupil of any size, this corresponds to a difference of roughly 16% in retinal irradiance. We do not expect these differences to play a major role in the effects we see here. Besides this, we assumed that rods do not contribute to circadian effects under photopic conditions. Thus, rhodopic effects were not constant across conditions (Supplementary Information, A and B). Data obtained in mice lacking functional cones or mice in which rods were the only photoreceptors suggest that rods support circadian photoentrainment even at higher light levels (that is, 500 lux)[61], although the effects on the SCN may be rather small[62]. Recent data suggest that the same holds true for humans living with functional achromatopsia, that is, individuals lacking functional cones[63]. However, the relative contributions of rod and melanopsin contributions are unknown, and some degree of adaptation of the circadian system in this congenital condition is likely[64,65]. Future studies should carefully disentangle the unique contributions of the rods to circadian photoreception across light levels. Finally, the use of a colour-calibrated stimulus flickering at 1 Hz hampers the generalizability of the results to real-life situations. While these conditions were chosen to specifically stimulate the post-receptoral S − (L + M) channel while keeping melanopic effects constant, they do not correspond to naturalistic light exposures. Importantly, constant light biased towards either pole of the post-receptoral S − (L + M) channel should in any case be less effective than the stimuli used here due to adaptation of cones and post-receptoral mechanisms under unchanging light conditions[27–34,66].

In summary, we found no conclusive evidence for an effect of calibrated changes in light colour along the blue–yellow axis at constant melanopic excitation on the human circadian system, psychomotor vigilance, sleepiness or sleep (that is, latency to 10 min of continuous sleep, SWA during the first sleep cycle) for a 1-h light exposure starting 30 min after HBT. This seems to underscore the primary role of melanopsin-containing ipRGCs in mediating these effects that have repeatedly been reported in the literature. From a more practical perspective, it seems that the human circadian clock

is relatively insensitive to shifts in light colour towards warmer colour temperatures at constant melanopic excitation. Smartphones and other displays with night-shift modes typically change colour and reduce melanopic excitation in a yoked fashion, and our study provides evidence that any effects seen in night-shift mode may be due to the reduction in melanopic excitation. As a large body of literature convincingly suggests that short-wavelength proportions of light should be reduced in the evening[37] to prevent decreases in sleepiness and a phase delay[37,43,67,68], we therefore encourage users of devices with background-lit displays (that is, smartphones, tablets and computer screens) to make use of built-in software or apps such as f.lux[69] in the evenings and during the night. In the future, tech companies may also opt to use metameric light that allows to reduce short-wavelength proportions without a change in perceived colour. Recently, Schöllhorn and colleagues[44] showed that low-melanopic light can indeed mitigate the unwanted effects of screen use at night, confirming the primary role of melanopsin photoreception in setting our circadian system by light.

## Methods
### Ethical approval
Approval for this study was granted from the Ethikkommission Nordwest- und Zentralschweiz with approval number 2020–02037. The study was conducted in accordance with Swiss law and the Declaration of Helsinki. It was registered as a clinical trial (DRKS00023603).

### Participants
Healthy, young male and female participants (age range 18–35 years; age range restricted to minimize age-related inter-observer variability in pre-receptoral filtering[72,73]) were recruited, with the goal of including an equal number of men and women in the study. Inclusion criteria were a body mass index between 18.5 and 24.9 kg m$^{-2}$ (that is, normal weight according to World Health Organization criteria, calculated from self-reported height and weight), an informed consent as documented by signature of the participant, and approval of study participation by the study physician. Participants were excluded if they were pregnant (self-report), suffered from chronic or debilitating medical conditions (physical examination by the study physician), used medications impacting on visual, neuroendocrine, sleep and circadian physiology (determined by study physician), used drugs (verified with a urine multi-drug screen during each experimental night; nal von minden GmbH), had performed shift work <3 months before beginning of the study, travelled across more than two time zones <1 month before beginning of the study, were characterized by an extreme chronotype according to the Munich Chronotype Questionnaire[74] (exclude values ≤2 or ≥7, include >2 and <7), had extremely short or long sleep duration (subjective sleep duration on workdays outside 6–10 h according to the Munich Chronotype Questionnaire), had a sleep efficiency <70% during the adaptation night (calculated as SEFF = total sleep time/ time in bed using visual scoring), any indicators of a sleep disorder (self-report or during the adaptation night) or photosensitive epilepsy. Furthermore, they were excluded in case of previous enrolment in this study or if they were the investigators' family members, employees or other dependent persons. They were also excluded if they were unable to understand and/or follow the study procedures or in case of non-compliance to sleep–wake times during the ambulatory parts. Participants were furthermore screened for normal colour vision using the Cambridge Colour Test[75] (trivector version) implemented using an iMac-based Metropsis system (Cambridge Research Systems), and were excluded if they did not have normal colour vision. Participants whose bedtime or wake time deviated by more than 30 min more than twice during the 5 days preceding each experimental visit were not empanelled in the study or excluded. All exclusion criteria are listed in Supplementary Table 23. Any exclusions were counted and reported. In case participants had to be excluded due to non-adherence to sleep–wake times during the circadian stabilization period, but had already

completed more than one study visit, none of their data were used. Two participants were planned to be tested at the same time (for more information, see 'Deviations from protocol' section). Participants received a remuneration of CHF 1000 after successful completion of the study protocol. The authors affirm that the depicted human provided written informed consent for publication of the images in Supplementary Picture 1.

## Protocol

In this within-subject protocol, all participants underwent the same study procedures, which started with an initial 7-day circadian stabilization period in which they were instructed to go to bed within 1 h (±30 min) of a target bedtime and rise within 1 h (±30 min) of a target wake-up time with a target sleep duration of 8 h. Both target bedtimes and wake-up times were agreed upon with the participants to match their HBT and ensure compatibility with daily life. Compliance to the individual sleep–wake schedules was ensured using wrist actimetry (Centre Point Insight Watch; ActiGraph LLC) and sleep logs[76]. They also had to adhere to this stabilization protocol during the ambulatory phases between further visits to the laboratory. In total, participants entered the laboratory four times. The first laboratory visit was an adaptation night, and the second, third and fourth laboratory visits were experimental visits both including two evenings, one night and one day (32 h (was actually 32.5 h; see 'Deviations from protocol' section)) in the lab (Fig. 2).

During the first night of each of the three experimental visits, participants were exposed to the constant light (that is, 'background' condition), a yellow-bright flickering condition, or a blue-dim flickering condition between 30 min and 90 min after HBT (1 h exposure duration). While the first visit was always the constant light condition ('background'), the order of the other two conditions was counterbalanced among participants and within each gender subgroup. Note that in the Stage 1 protocol it was incorrectly stated that the order of all three conditions would be randomized, which deviated from the information provided in Fig. 2, its caption and the hypotheses. Compliance with instructions to keep eyes open during the light exposure was checked informally through electrooculography, which had to show blinks. Experimental visits were spaced by a wash-out phase of 1 week (that is, five nights).

On each experimental visit, participants entered the laboratory 6.5 h before their HBT. Upon arrival, a small dinner was served and the EEG, electrooculogram (EOG) and electromyogram (EMG) for later sleep assessment (that is, PSG, see below) was placed. From 5 h before HBT until 1 h 30 min after HBT, participants provided saliva samples every 30 min, using Salivettes (Sarstedt), which were centrifuged at 1,448$g$ for 3 min and frozen at −20° (was actually −28 °C, see 'Deviations from protocol' section) for later assaying.

Participants rated subjective sleepiness on the KSS[77], completed a modified auditory PVT (6 min; was actually 10 min (see 'Deviations from protocol' section)) to assess behavioural vigilance, and rated their visual comfort with every melatonin sample. We used the German version of the KSS ('Bitte bewerten Sie Ihre Müdigkeit': 'sehr wach' (1), 'wach' (3), 'weder wach noch müde' (5), 'müde, aber keine Probleme, wach zu bleiben' (7), 'sehr müde, große Probleme, wach zu bleiben, mit dem Schlaf kämpfend' (9)).

Furthermore, we recorded a 3-min resting-state EEG (open eyes) just before the start of the light exposure, 30 min into light exposure, and immediately after light exposure (that is, 27 min, 1 h and 1 h 27 min after HBT). Following a visit to the bathroom (approximately 15 min), they went to bed 2 h after their HBT for a 6-h sleep opportunity. Following wake-up they provided another five saliva samples during the first 2 h of wakefulness. They then spent the day in the lab under controlled lighting conditions (<8 lux; fluorescent lighting), and we served small meals every 2.5 h (five snacks with 0.2 of the total basic metabolic rate estimated with the Mifflin–St. Jeor equation[78]; was actually six snacks,

see 'Deviations from protocol' section). In the second evening, we again started sampling melatonin in 30-min intervals 5 h before HBT and obtained melatonin samples, KSS ratings and PVT measurements until 1 h 30 min after HBT. For an illustration of the study protocol, please see Fig. 2. Please note that there were no hypotheses for the KSS or PVT in the second evening, but the protocol was the same on both evenings. They then went to bed for a 9-h recovery sleep opportunity. The protocol formally ended 1 h 30 min after HBT in the second night; however, participants were offered to sleep in the lab. Throughout light exposure, participants' compliance was be monitored using infra-red cameras pointed at the participants.

Throughout the repeated laboratory visits, participants had no knowledge of external time and were not allowed to use their phones (except in emergencies) or laptops. They were allowed to read magazines, books or other material (e-readers were allowed if the corneal illuminance was <8 lux). Small gaming devices such as GameBoys (monochrome display) or table-top and card games were provided. Participants were allowed to listen to podcasts provided they had a device that did not tell the time.

**Adaptation night.** Participants came for an initial adaptation night (8-h sleep opportunity) to the laboratory to screen for potential sleep disorders (see below for the PSG setup) and to make them acquainted with the laboratory setting. Participants showing signs of sleep disorders or presenting with sleep efficiency <70% would have been excluded from the study.

**PSG.** For resting-state and nocturnal PSG recordings (that is, EEG, EOG and EMG) we used ambulatory BrainProducts LiveAmp devices. We planned to record PSG at a sampling rate of 250 Hz (was actually 500 Hz; see 'Deviations from protocol' section) from 21 scalp channels mounted in a cap (EasyCap) and 4 EOG channels with FCz as the online reference. In addition, we placed two mastoid electrodes for later re-referencing according to the sleep staging criteria of the American Association of Sleep Medicine, as well as two chin EMG electrodes and two electrocardiogram electrodes[79]. During the adaptation nights, we additionally recorded from two EMG electrodes on the tibialis anterior muscle of one leg to screen for nocturnal leg movements, a respiration belt and a nasal cannula to screen for respiratory problems.

**Visual comfort questionnaire.** Participants were planned to be asked to rate or respond to various aspects of the light exposure using a six-question, seven-item Likert scale questionnaire[12]. This questionnaire was administered in German. The questions were planned to be about the comfort of light ('Allgemein ist das Licht angenehm'; überhaupt nicht (1) to sehr stark (7)), the perceived brightness ('Wie empfinden Sie die Helligkeit des Lichtes?'; sehr dunkel (1) to sehr hell (7)), light level preference ('Ich hätte es lieber …'; deutlich dunkler (1) to deutlich heller (7)), glare ('Dieses Licht blendet mich'; überhaupt nicht (1) to sehr stark (7)), the perceived colour temperature ('Wie empfinden Sie die Lichtfarbe?'; sehr kalt (1) to sehr warm (7)) and general wellbeing ('Wie fühlen Sie sich im Moment?'; unwohl (1) to wohl (7)). For deviations from the planned questionnaire, please see the 'Deviations from protocol' section.

## Light exposure and stimuli

**Evening light stimuli.** Light exposure was planned to be delivered from a vertical front lighting panel (width 220 cm, height 140 cm) that consists of 24 light-emitting diode (LED) panels (RGBW), each containing 144 LEDs (that is, total of 3,456 LEDs) covered by diffusing material[80,81]. Two participants were planned to sit next to each other at a distance of 1.5 m from the wall, facing the wall. The LED primaries have the following properties (peak wavelength ± full width at half maximum; CIE 1931 $xy$ chromaticity): blue (462 ± 23 nm; 0.14, 0.06), green (518 ± 32 nm; 0.16, 0.72), red (631 ± 16 nm; 0.70, 0.30) and white (peaks 446 nm and 560 nm; 0.33, 0.34). Due to the coronavirus pandemic, an alternative

setup with two separate laboratories had to be used. For details, please see the 'Deviations from protocol' section.

The wall was be calibrated from a vantage point centred between the two participants, and both irradiance and radiance measurements were planned to be taken. To characterize spectral shifts of the LED wall, we planned to measure the spectrum of each of the RGBW primaries at 16 settings at 8-bit resolution.

Stimuli were either a constant light matched in its photoreceptor activation profile to that of daylight at 6,500 K (D65, used during the first experimental visit; 'background condition'; Fig. 1f (note that the Stage 1 protocol wrongly stated that this light was used during the adaptation night and the second night of each experimental visit)), or sinusoidally flickering light stimulating the S cones in an balanced but opponent fashion with luminance (blue-dim: planned +37% S-cone contrast, planned −37% L + M contrast from background, Fig. 1g; yellow-bright: planned −37% S-cone contrast; planned +37% L + M cone contrast from background, Fig. 1h), alternating flickering for 30 s to constant background for 30 s. The flicker frequency was 1 Hz so as to provide a continuous tonic signal to the cones that might otherwise adapt to a continuous light biased towards either −S + (L + M) or +S − (L + M) (Fig. 1e). This frequency was also used in the habituation stimulus in a seminal study determining the cardinal directions of colour space[18]. To avoid adaptation to flicker, we flickered for 30 s, then held the light constant at background for 30 s, corresponding to the approximate time scale of habituation in the purported S-opponent psychophysical channel[18]. All stimulus conditions are summarized in terms of their nominal photoreceptor excitation[82] in Table 1.

To find the settings on the RGBW channels of the lighting panel, we implemented an optimization minimizing the squared error between desired and actual contrasts using 'fmincon' routine as implemented in MATLAB (MathWorks). Critically, our stimuli were designed to produce no differential stimulation of melanopsin-expressing ipRGCs (0%).

**Light during the scheduled day.** During the scheduled day (that is, from wake-up until the end of the protocol), participants were planned to be in dim light (<8 lux) provided by fluorescent lighting.

**Data collection and processing**
All data handling was done in R (ref. 83), except for EEG data. EEG data processing was done in MATLAB (MathWorks) using the 'Fieldtrip' toolbox for MATLAB[84] first before the results were exported for statistical analyses in R.

**Hormone concentrations (melatonin).** Analyses of salivary melatonin were planned to be done with enzyme-linked immunoabsorbent assays (ELISAs) in an in-house laboratory. We planned to use ELISA kits (Novo-LytiX), which have a minimal detection limit (limit of quantification) of at least 1.6 pg ml$^{-1}$. Eventually, a RIA was used instead of an ELISA (see 'Deviations from protocol' section). The DLMO was determined by fitting evening melatonin profiles by a piecewise linear-parabolic function using the hockey-stick algorithm (planned: v2.4, used version: 2.5; see 'Deviations from protocol' section) to calculate the DLMO[85].

**Control variables.** For the PVT data, trials with RTs ≥100 ms were considered valid trials[39]. We then computed the median RTs as well as RTs for the fastest and slowest deciles of valid trials. KSS ratings were analysed without further pre-processing steps. As for the neuroendocrine responses, statistical evaluation was planned to be with repeated-measures ANOVAs and planned contrasts (note that we implemented an ANOVA-like approach using linear models to circumvent case-wise deletion; see 'Deviations from protocol' section).

**EEG analyses.** Following data acquisition, the signal from EEG channels was filtered and re-referenced to a linked mastoids reference. For the analyses of objective sleepiness, we first corrected for eye

movements using an independent component analysis, which was followed by manual exclusion of other artefacts (for example, muscle artefacts). Subsequently, artefact-free data were segmented into 2-s time bins and subjected to fast-Fourier transformations (FFT) using the 'mtmfft' function as implemented in the Fieldtrip toolbox yielding a frequency resolution of 0.5 Hz. We then extracted absolute power in the theta (4–7 Hz) and alpha (8–12 Hz) frequency range and averaged across parietal and occipital electrodes (that is, P3, Pz, P4, O1, Oz and O2) and segments within each frequency band. We then computed the alpha/theta ratio. For sleep analyses, we had planned to score sleep semi-automatically with an algorithm (The SIESTA Group[86,87]). These analyses were planned to be based on EEG data from electrodes F3, F4, C3, C4, O1 and O2 (downsampled to 128 Hz and re-referenced to the contralateral mastoid electrode) along with the signal from the EOG channels that have been placed according to Rechtschaffen and Kales criteria and the chin EMG signal. However, for budgetary reasons, we eventually used the Somnolyzer algorithm as implemented in the Philips Respironics Sleepware G3 software instead of the SIESTA scoring (see 'Deviations from protocol' section). Additionally, we computed EEG SWA (that is, delta power density between 0.5 Hz and 4.5 Hz) as an indicator of sleep propensity across the night within each NREM part of a sleep cycle. To this end, we computed EEG SWA for each decile of the NREM part of the first NREM-REM cycle[88,89]. For the computation of delta power density, artefact-free data were segmented into 2-s time bins and subjected to FFTs yielding a frequency resolution of 0.5 Hz. Thereafter, FFT results were averaged in the 0.5–4.5 Hz range at frontal electrodes F3, Fz and F4 within each percentile of each NREM cycle. For each NREM cycle, the analyses thus yielded ten measures per participant. The analysis procedure described here is based on a publication by Chellappa and colleagues[25].

**Statistical data analysis**
All statistical analyses were performed in R (ref. 83). Statistical tests were performed with the BayesFactor package[90,91], and we planned to implement an ANOVA design (function 'anovaBF'). Note that we implemented an ANOVA-like approach using linear models instead of the classic ANOVA function to circumvent case-wise deletion in case of missing data, for details see the 'Deviations from protocol' section. The analytic strategy is described in Supplementary Table 2. Participant ID and gender were entered as random factors.

**Exclusion criterion.** Only data from participants for whom the light exposure took place during the rising arm of the melatonin curve and after the DLMO were included in the analyses.

**Bayesian sampling strategy using sequential Bayes factors.** Our sample size and recruitment approach were intimately linked to the analytic strategy. We intended to collect data until obtaining sufficient evidential strength for our primary hypothesis (C1) that the circadian phase shift is larger in the yellow-bright flickering condition than in the blue-dim flickering condition and the constant light condition, or until reaching our resource limit (n = 16). Formally, this would have been achieved by calculating sequential Bayes factors[35]. First, the repeated-measures ANOVA described above would have been run sequentially after data from each new participant was included in the data set. Participants would then have continuously been recruited for the study until sufficient evidence for the full model would have been reached (male, female alternating). We consider a BF of 10 as 'strong' evidence, following standard categorizations: 1 < BF < 3—anecdotal evidence, 3 < BF < 10—moderate, 10 < BF < 30 to BF >30—very strong evidence. According to the sequential Bayes factor approach, if this threshold was reached for the primary hypothesis, the study would have been halted, and the evidence strength would have been reported according to the Bayes factor. Otherwise, the study would have been halted if the resource limit is reached (n = 16). A minimum number of

four participants (two men, two women) were planned to participate in the study, in case the evidence threshold would have been reached with one, two or three participants already.

### Deviations from protocol

**Protocol.** The protocol lasted for 32.5 h instead of the 32 h previously described in the text, because we initially did not take into account that the last melatonin sample and round of assessments would take approximately 15–20 min.

Saliva samples were frozen at −28 °C rather than the previously stated −20 °C, because we used a different freezer.

Instead of a 6-min version of the PVT, we used a 10-min version. Specifically, the 10-min version has been associated with increased sensitivity to modulations of both sleep homeostatic and circadian drives and to improvements in alertness after wake-promoting interventions[39]. A shorter version has only been recommended if the protocol does not allow for the 10-min version[92,93]. As the protocol allowed us to use the longer version, we felt using the longer version would improve the validity of our measurements.

Instead of five snacks every 2.5 h with 0.2 of the total basic metabolic rate estimated with the Mifflin–St. Jeor equation[78], we served six snacks of 0.17 of the total basic metabolic rate every 2.5 h. This was to shorten the time between the last snack and the bedtime to 5.5 instead of 8 h, and thus to prevent participants from becoming too hungry in the evening. Additionally, we allowed participants to use their phones during a 'social hour' scheduled 4 h after wake-up as we have previously experienced this to increase adherence to long study protocols. Importantly, participants wore orange-tinted blue-blocking glasses during this time and were instructed to keep display brightness as low as possible.

**PSG.** We recorded data at a sampling rate of 500 Hz instead of the previously planned 250 Hz with 500 Hz being the standard sampling rate in our laboratory. However, as we were interested only in frequencies up to 12 Hz, this change has not affected the analyses or results.

**Visual comfort questionnaire.** Due to internal communication issues, the visual comfort questionnaire slightly deviated from the planned version and corresponded to a version used in an earlier study conducted in the laboratory. In more detail, instead of a seven-item Likert scale, a five-item Likert scale was used. Furthermore, the questions slightly deviated from the initially planned ones. Importantly though, the relevant questions to assess visual comfort were still included. Contrary to what had been planned, participants were not asked to rate their light level preference ('I would prefer it to be lighter/darker') and their general wellbeing ('I feel well/unwell'). Instead, participants rated the light according to how wake-promoting versus tiring it was ('Die Beleuchtung in diesem Zimmer…'; hilft mir, wach zu bleiben (1) to ermüdet mich stark (5); Engl.: 'The lighting in this room…'; helps me to stay awake (1) to is very tiring (5)) and whether they felt it helped them to concentrate ('Die Beleuchtung in diesem Zimmer…'; hilft mir, mich besser zu konzentrieren (1) to stört meine Konzentrationsfähigkeit (5); Engl.: 'The lighting in this room…'; helps me to concentrate (1) to disturbs my ability to concentrate (5)). These two questions were however not (planned to be) included in the calculation of the visual comfort score; the relevant questions were still asked.

**Light exposure and stimuli.** Due to the coronavirus pandemic, it was not feasible to have two participants sit in front of the planned LED wall for several hours next to each other. Thus, we used two custom-made displays, which provided a stimulus that was functionally equivalent to the originally planned stimulus, in two separate comparable (sleep) laboratories. Specifically, light exposure delivered through a modified 27-inch display consisting of a total of five sets of primaries, with peak spectral emissions at 430, 480, 500, 550 and 630 nm (refs. 44,94). The backlights were independently controlled at 8-bit resolution using

Digital Multiplex (DMX). For an illustration of the laboratory setup, please see Supplementary Picture 1.

Due to some technical differences in the device primaries between the planned LED wall and the displays, the validated contrast between the 'blue-dim' and the 'yellow-bright' conditions was ±25% instead of the originally planned nominal contrast of ±37% (blue-dim: +25% S-cone contrast, −25% L + M contrast from background, Fig. 1g; yellow-bright: −25% S-cone contrast; +25% L + M cone contrast from background, Fig. 1h). This validated contrast (±25%), although lower, however still yields a psychophysically substantial and physiologically meaningful contrast on the post-receptoral channel.

**Light during the scheduled day.** Sitting accommodations and furniture were placed in a way to prevent illuminances higher than 8 lux, and, if necessary, participants were instructed to not take positions in the room associated with higher illuminance levels. Nevertheless, for short periods of time, illuminances may have exceeded 8 lux (Supplementary Table 6).

**Hormone concentrations (melatonin).** Analyses of salivary melatonin were done with radioimmunoassay (RIA) rather than ELISAs in the NovoLytiX laboratory rather than an in-house laboratory. RIA and ELISAs by NovoLytiX generally reveal comparable results, but the laboratory uses RIA as analyses are less time consuming. The RIA has an analytical sensitivity of 0.2 pg ml⁻¹ and a limit of quantification of 0.9 pg ml⁻¹. For the calculation of the DLMO, we used version 2.5 of the hockey-stick algorithm (planned: v2.4), which was the most recent one available at the time of the data analysis[85].

**EEG analyses.** Unlike originally planned we did not use the service provided by the SIESTA group for budgetary reasons. Instead, sleep was scored automatically with the Somnolyzer algorithm as implemented in the Philips Respironics Sleepware G3 software v. 4.0.1.0 (refs. 86,87). Importantly, the Somnolyzer algorithm is based on the algorithm used by the SIESTA group and an automatic scoring software that has been cleared by the U.S. Food and Drug Administration (FDA). Due to the change of the algorithm, analyses also included data from electrodes P3 and P4 in addition to F3, F4, C3, C4, O1 and O2. Furthermore, the data did not have to be downsampled to 128 Hz before analysis. However, the G3 software required us to apply some filtering before the sleep stage analyses. Filtering was according to AASM criteria[95] with the EEG and EOG signals being filtered between 0.3 Hz and 35 Hz and the EMG between 10 Hz and 100 Hz with a Notch filter at 50 Hz to remove line noise.

**Statistical data analyses.** Instead of a classic ANOVA design using the function 'anovaBF', we implemented an ANOVA-like design using linear models and the function 'lmBF'. We chose to use linear models instead of the classic ANOVA function as this would have resulted in a case-wise deletion in case of missing data (for example, if data from even just 1 out of 14 PVTs per evening were not available or one DLMO out of the three conditions could not be determined).

### Protocol registration

The approved Stage 1 protocol can be found at ref. 70, and a table of the laboratory protocol can be found at ref. 71.

### Reporting summary

Further information on research design is available in the Nature Portfolio Reporting Summary linked to this article.

## Data availability

All data generated in this study (including laboratory logs) are available in anonymized and de-identified form on FigShare (data: https://doi.org/10.6084/m9.figshare.23578698 ref. 96; laboratory log: https://doi.org/10.6084/m9.figshare.23578695).

## Code availability

Code to simulate and analyse data and implement statistical models is available on a dedicated GitHub repository (https://github.com/ChristineBlume/Effects-of-calibrated-blue-yellow-changes-in-light-on-the-human-circadian-clock/), which will be archived as a snapshot on FigShare. The simulation code was shared with reviewers via the journal in the Stage 1 submission and was be made public at Stage 2.

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

## Acknowledgements

We thank the funders that enabled this research. The funders had no role in study design, data collection and analysis, decision to publish or preparation of the manuscript. Specifically, C.B. was supported by a scholarship from the University of Basel's research fund for excellent early career researchers as well as an Ambizione grant from the Swiss National Science Foundation awarded to her. Additionally, the study was funded by a grant from the Forschungsförderungsfonds of the Psychiatric Hospital of the University of Basel (UPK) awarded to M.S. and funding for Scientific Exchange by the Swiss National Science foundation awarded to M.S. and C.C. During parts of this work, M.S. was supported by a Sir Henry Wellcome Postdoctoral Fellowship (Wellcome Trust, 204686/Z/16/Z) and Linacre College, University of Oxford (Biomedical Sciences Junior Research Fellowship). Additionally, we thank all participants, who volunteered to participate in the project. The data acquisition has mainly been done by an amazing team of helping hands, mainly interns, and master students. We are incredibly grateful to Z. Butt, who was also the study coordinator, S. Bogazlyianlioglu, L. Fricke, A. von Gatterburg, Y. Hao, B. Knezevic, A.-S. Loock, I. Messina, J. Miller, M. Vettiger, L. Vilela, P. Weiss, K. Wieczorek, R. Zeugin and T. Zumbrunn. Thank you for your team spirit, your flexibility and for making this project possible. We also thank F. Fazlali for her help with the calibration of the display, M. Münch for sharing her knowledge on the determination of DLMOs using the Hockeystick algorithm and J. Weber from NovoLytiX for constant support with the melatonin assays. Last, we also thank the study physicians M. P. Meyer, and C. Epple as well as M. Cattaneo, statistician at the Department of Clinical Research of the University of Basel, for their valuable support and advice.

## Author contributions

C.B. and M.S. conceptualized the project and wrote the Stage 1 Registered Report supported by C.C. C.B., M.S. and C.C. acquired the funding. C.B. acquired the data, conducted the analyses and wrote the first draft of the full manuscript supported by M.S. I.S. and H.C.S. greatly supported the data acquisition. All authors provided a critical review of the manuscript and approved the submitted version.

## Funding

## Competing interests

Related to lighting, M.S. is currently an unpaid member of CIE Technical Committee TC 1–98 ('A Roadmap Toward Basing CIE Colorimetry on Cone Fundamentals'). M.S. was an unpaid advisor to the Division Reportership DR 6–45 of Division 3 ('Publication and maintenance of the CIE S026 Toolbox') and a member of the CIE Joint Technical Committee 9 on the definition of CIE S 026:2018. Since 2020, M.S. is an elected Member of the Daylight Academy and an unpaid member of the Board of Advisors of the Center for Environmental Therapeutics. C.C. has had the following commercial interests related to lighting: honoraria, travel, accommodation and/ or meals for invited keynote lectures, conference presentations or teaching from Toshiba Materials, Velux, Firalux, Lighting Europe, Electrosuisse, Novartis, Roche, Elite, Servier and WIR Bank. C.C. is an elected member of the Daylight Academy. C.B. has had the following commercial interests related to sleep and/or light: honoraria for invited talks and workshops from IKEA, F.A. Hoffmann-La Roche AG, L'Oréal and Vattenfall. C.B. is an elected member of the Daylight Academy. The remaining authors declare no competing interests.

## Additional information

**Correspondence and requests for materials** should be addressed to Christine Blume or Manuel Spitschan.

Prof. Manuel Spitschan

# Reporting Summary

## Statistics

For all statistical analyses, confirm that the following items are present in the figure legend, table legend, main text, or Methods section.

| n/a | Confirmed | |
|---|---|---|
| ☐ | ☒ | The exact sample size (*n*) for each experimental group/condition, given as a discrete number and unit of measurement |
| ☐ | ☒ | A statement on whether measurements were taken from distinct samples or whether the same sample was measured repeatedly |
| ☐ | ☒ | The statistical test(s) used AND whether they are one- or two-sided<br>*Only common tests should be described solely by name; describe more complex techniques in the Methods section.* |
| ☐ | ☒ | A description of all covariates tested |
| ☐ | ☒ | A description of any assumptions or corrections, such as tests of normality and adjustment for multiple comparisons |
| ☐ | ☒ | A full description of the statistical parameters including central tendency (e.g. means) or other basic estimates (e.g. regression coefficient) AND variation (e.g. standard deviation) or associated estimates of uncertainty (e.g. confidence intervals) |
| ☒ | ☐ | For null hypothesis testing, the test statistic (e.g. *F*, *t*, *r*) with confidence intervals, effect sizes, degrees of freedom and *P* value noted<br>*Give P values as exact values whenever suitable.* |
| ☐ | ☒ | For Bayesian analysis, information on the choice of priors and Markov chain Monte Carlo settings |
| ☒ | ☐ | For hierarchical and complex designs, identification of the appropriate level for tests and full reporting of outcomes |
| ☒ | ☐ | Estimates of effect sizes (e.g. Cohen's *d*, Pearson's *r*), indicating how they were calculated |

*Our web collection on statistics for biologists contains articles on many of the points above.*

## Software and code

Policy information about availability of computer code

| Data collection | - Screening for normal colour vision: Cambridge Colour Test46 (trivector version) implemented using an iMac-based Metropsis system (Cambridge Research Systems, Rochester, UK)<br>- EEG data collection: BrainVision Recorder Software (BrainProducts GmbH, Gilching, Germany)<br>- Psychomotor Vigilance Task (PVT) and administration of questionnaires (i.e., Karolinska Sleepiness Scale [KSS], visual comfort): Python version 3.6 (Python Software Foundation) using PsychoPy version 3.1.5.<br>- Control of the LEDs in the custom-made display: Q Light Controller + software for DMX control (https://www.qlcplus.org/) |
|---|---|
| Data analysis | - EEG analyses: EEG raw data were analysed using the Fieldtrip toolbox (Oostenveld et al., 2010; distribution from https://gitlab.com/obob/obob_ownft) running on MATLAB 2022a (The Mathworks, Natick, MA, USA).<br>- Sleep staging: Philips Respironics Sleepware G3 software version 4.0.1.0.<br>- Dim Light Melatonin Onset: Hockey-stick algorithm version 2.5 (cf. Danilenko et al., 2014)<br>- Statistical analyses: R version 4.2.3 using the BayesFactor package (all analysis codes are available here https://github.com/ChristineBlume/Effects-of-calibrated-blue-yellow-changes-in-light-on-the-human-circadian-clock/) |

For manuscripts utilizing custom algorithms or software that are central to the research but not yet described in published literature, software must be made available to editors and reviewers. We strongly encourage code deposition in a community repository (e.g. GitHub). See the Nature Portfolio guidelines for submitting code & software for further information.

# Data

Policy information about availability of data

All manuscripts must include a data availability statement. This statement should provide the following information, where applicable:
- Accession codes, unique identifiers, or web links for publicly available datasets
- A description of any restrictions on data availability
- For clinical datasets or third party data, please ensure that the statement adheres to our policy

All data generated in this study (including laboratory logs) are available in anonymised and deidentified form on FigShare (Data: https://doi.org/10.6084/m9.figshare.23578698; Laboratory Log: https://doi.org/10.6084/m9.figshare.23578695)

# Research involving human participants, their data, or biological material

Policy information about studies with human participants or human data. See also policy information about sex, gender (identity/presentation), and sexual orientation and race, ethnicity and racism.

| | |
|---|---|
| Reporting on sex and gender | Sex was determined based on self-reporting. Note that in German, there is only one word for sex/gender ("Geschlecht"), which is commonly interpreted as "sex". We collected data from an equal number of men and women and sex was a control variable (i.e., random effect, termed "gender" in the codes as it was self-reported) in our analyses. In the data that we publish with this manuscript, sex is included at the participant level (m = male vs. f = female). |
| Reporting on race, ethnicity, or other socially relevant groupings | As we did not expect social variables, race or ethnicity to affect our results, we did not have exclusion criteria relating to such variables. Thus, we do not expect that there was a relevant social grouping effect and we have no record of such variables. The only control variable that was included in the analyses was sex. |
| Population characteristics | Participants were young and healthy both mentally and physically (cf. exclusion criteria). The age range was limited to 18-35, which resulted in a mean age of 25.5±2.7 years. An equal number of men and women was included in the study. |
| Recruitment | Participants were recruited through an ad on a website run by the University of Basel, which includes a job board (www.markt.unibas.ch). |
| Ethics oversight | Approval for this study was granted from the Ethikkommission Nordwest- und Zentralschweiz (EKNZ) with approval number 2020-02037. |

Note that full information on the approval of the study protocol must also be provided in the manuscript.

# Field-specific reporting

Please select the one below that is the best fit for your research. If you are not sure, read the appropriate sections before making your selection.

☒ Life sciences        ☐ Behavioural & social sciences        ☐ Ecological, evolutionary & environmental sciences

For a reference copy of the document with all sections, see nature.com/documents/nr-reporting-summary-flat.pdf

# Life sciences study design

All studies must disclose on these points even when the disclosure is negative.

| | |
|---|---|
| Sample size | Assuming a large effect size (ES; Cohen's d = 0.8) and that H1 better predicts the data than H0, Bayes Factor Design simulations revealed that with 16 participants 62% of the simulations showed evidence for H1 with 38% being inconclusive (medium ES: 22.2% vs. 77.7%; small ES: 0.5% vs. 96.4% and 3.1% showing evidence for H0). However, we argue that, if the S–(L+M) opponent system indeed exerts a very strong effect on the circadian system, then this should be visible on a single-subject level already. In fact, previous research suggests that the assumptions about effect sizes (i.e., d = 0.8) for the circadian phase shifts used in the simulations outlined above may be rather conservative. Our financially driven resource limit was thus n=16 participants. |
| Data exclusions | No data were excluded from the analyses. |
| Replication | There was no internal replication. However, we have provided as much detail as possible and necessary to replicate the experiment. |
| Randomization | All participants underwent all study conditions. The order of the conditions was partly randomised with two possible orders. Participants were allocated to the order by participant number (even vs. uneven number). |
| Blinding | Blinding was not possible, because the experimenters had to select the correct light exposure and the light exposure conditions were visually distinguishable. |

# Reporting for specific materials, systems and methods

We require information from authors about some types of materials, experimental systems and methods used in many studies. Here, indicate whether each material, system or method listed is relevant to your study. If you are not sure if a list item applies to your research, read the appropriate section before selecting a response.

## Materials & experimental systems

| n/a | Involved in the study |
|-----|----------------------|
| ☒ ☐ | Antibodies |
| ☒ ☐ | Eukaryotic cell lines |
| ☒ ☐ | Palaeontology and archaeology |
| ☒ ☐ | Animals and other organisms |
| ☐ ☒ | Clinical data |
| ☒ ☐ | Dual use research of concern |
| ☒ ☐ | Plants |

## Methods

| n/a | Involved in the study |
|-----|----------------------|
| ☒ ☐ | ChIP-seq |
| ☒ ☐ | Flow cytometry |
| ☒ ☐ | MRI-based neuroimaging |

## Clinical data

Policy information about clinical studies

All manuscripts should comply with the ICMJE guidelines for publication of clinical research and a completed CONSORT checklist must be included with all submissions.

| | |
|---|---|
| Clinical trial registration | DRKS00023603 |
| Study protocol | The approved Stage 1 protocol can be found here (https://springernature.figshare.com/articles/journal_contribution/Effects_of_calibrated_blue-yellow_S_L_M_S_L_M_changes_in_light_on_the_human_circadian_clock_Registered_Report_Stage_1_Protocol_/13050215). For any deviations from this original protocol, please see the "Deviations from Protocol" section in the manuscript. A table of the laboratory protocol can be found here (https://figshare.com/articles/dataset/Protocol/23578704). |
| Data collection | Data acquisition took place continuously between March and December 2022 at the facilities of the Centre for Chronobiology of the University of Basel with a break of 4 weeks in August 2022 (for more details on the distribution of participants across the acquisition period incl. subjectively reported light history on the day of the experimental visit, please see the supplemental material S3 and the laboratory log). Participants always entered the lab on the same day of the week. |
| Outcomes | The primary outcome measure was differences in dim light melatonin onset (DLMO) between evenings 1 and 2 of each experimental visit. The DLMO was assessed using the Hockeystick method (version 2.5; Danilenko et al., 2014).

The following secondary outcome measures were assessed:
- Melatonin concentrations were assessed with saliva samples every 30 min starting 5 hours prior to habitual bedtime (HBT) until 1h 30 min after HBT. The samples were assayed by Novolytix GmbH (Pfeffingen, Switzerland) with radioimmunoassays (RIA).
- Subjective sleepiness was assessed with the Karolinska Sleepiness Scale every 30 min starting 5 hours prior to HBT until 1h 30 min after HBT (Akerstedt & Gilberg, 1990; Nordin et al., 2013).
- Objective sleepiness was assessed using EEG-derived alpha (8-12 Hz)/theta (4-7 Hz) ratios assessed during resting-state EEG measurements (3 min, eyes open) before, 30 min into, and after light exposure at electrodes P3, Pz, P4, O1, Oz, O2.
- Visual comfort was assessed using 5-point Likert scales. Participants rated visual comfort every 30 min starting 5 hours prior to HBT until 1h 30 min after HBT. Visual comfort was calculated as the average rating from the responses to the questions about how pleasant the room lighting was generally, how pleasant the brightness was, how glaring the artificial light was, and how pleasant participants rated the colour temperature.
- PVT measures comprised median reaction time, slowest 10% reaction times, and fastest 10% reaction times during a 10-min PVT that was administered every 30 min starting 5 hours prior to HBT until 1h 30 min after HBT. All RTs < 100 ms were dismissed as invalid trials.
- EEG-derived sleep onset latency to 10 min of continuous sleep were based on sleep staging with the Philips Respironics Sleepware G3 software v. 4.0.1.0.
- EEG-derived slow wave activity (SWA) during the first sleep cycle was assessed as the averaged power between 0.5 and 4.5 Hz at electrodes F3, F4, Fz. We further separated the first sleep cycles into percentiles using the "SleepCycle" package for R (Blume & Cajochen, 2021; https://doi.org/10.1016/j.mex.2021.101318).
- Brightness was assessed using a 5-point Likert scale. Participants rated perceived brightness every 30 min starting 5 hours prior to HBT until 1h 30 min after HBT. |

