## [Peer Review File · Nature Human Behaviour]

Peer Review Information

Journal: Nature Human Behaviour

Manuscript Title: Effects of calibrated blue–yellow changes in light on the human circadian clock

Corresponding author name(s): Christine Blume and Manuel Spitschan

Reviewer Comments & Decisions:

Decision Letter, initial version:

19th May 2020

Dear Manuel,

Thank you once again for your manuscript, entitled "Effects of calibrated blue-yellow ($-S+[L+M]$, $+S-[L+M]$) changes in light on the human circadian clock," and for your patience during the peer review process.

Your manuscript has now been evaluated by 3 reviewers, whose comments are included at the end of this letter. Although the reviewers find your protocol to be of interest, they also raise some important concerns. We are very interested in the possibility of proceeding further with your submission in Nature Human Behaviour, but would like to consider your response to these concerns in the form of a revised manuscript before we make a decision on in principle acceptance and Stage 2 submission.

To guide the scope of the revisions, the editors discuss the referee reports in detail within the team, including with the chief editor, with a view to (1) identifying key priorities that should be addressed in revision and (2) overruling referee requests that are deemed beyond the scope of the current study. We hope that you will find the prioritized set of referee points to be useful when revising your study. Please do not hesitate to get in touch if you would like to discuss these issues further.

First, we ask you to follow the recommendations made by Reviewer #2, that pertain to a) ensuring that participants are exposed to the different conditions (i.e. that your manipulation works), and b) that you include an outcome neutral measure, i.e. use a test that is not affected by different light conditions directly.

Second, because your Bayesian sampling strategy is not open, but guided by a feasibility criterion (i.e. you specify a maximum N) we ask you to include simulations (as specified in <https://osf.io/d4dcu/>) to show, depending on a range of effect whether your maximum N is likely to be sufficient to reach

strong conclusions on your predictions.

Moreover, we also ask you to take on the feedback offered by Reviewer #3, and update your exclusion criteria. All three reviewers make further helpful suggestions which we ask you to implement, in particular highlighting Reviewer #1's remarks regarding the motivation and clarity of the predictions.

We ask you to follow the attached template and checklist to the letter. This entails changing your Table 1 to the obligatory Design Table. Note that the information currently included in Table 1 in the columns "Hypothesis" and "Analysis" is underspecified for a Design Table. For example, for the Statistical Analysis, we would ask you to include a description of the statistical test (ANOVA, repeated-measures ANOVA), listing all factors and factor levels (the column is designated "Analysis Plan"). You must also include a column (designated "Interpretation given different outcomes") where you list how other patterns of results, that are not predicted on the basis of your hypothesis, would be interpreted. If you wish to interpret any evidence as refuting your hypothesis (rather than being inconclusive), you must do this on the basis of Bayes Factors, and this should be explained in this column.

Your current Table 1 includes other useful information that will not enter the Design Table. I would recommend placing this information into a new Table 2.

Finally, I would strongly encourage you to deposit the analysis script at this stage (and include it in your Code Availability statement), ideally together with simulated data. This considerably strengthens the transparency and replicability of Registered Reports, and offers authors an opportunity another chance to catch small mistakes early on.

In sum, we invite you to revise your Stage 1 Registered Report taking into account reviewer and editor comments. Please highlight all changes in the manuscript text file.

* Include a "Response to reviewers" document detailing, point-by-point, how you addressed each referee comment. If no action was taken to address a point, you must provide a compelling argument. This response will be sent back to the reviewers along with the revised manuscript.

* In your cover letter, please include the following information:

- An anticipated timeline for completing the study if your Stage 1 submission is accepted in principle.
- A statement confirming that you agree to share your raw data, any digital study materials, computer code (if relevant), and laboratory log for all eventually published results.
- A statement confirming that, following Stage 1 in principle acceptance, you agree to register your approved protocol on the Open Science Framework (<https://osf.io/>) or other recognised repository, either publicly or under private embargo, until submission of the Stage 2 manuscript.
- A statement confirming that if you later withdraw your paper, you agree to the Journal publishing a short summary of the pre-registered study under a section Withdrawn Registrations.

[REDACTED]

We hope to receive your revised manuscript within four to eight weeks. If you cannot send it within this time, please let us know. We will be happy to consider your revision so long as nothing similar has been accepted for publication at Nature Human Behaviour or published elsewhere.

Nature Human Behaviour is committed to improving transparency in authorship. As part of our efforts in this direction, we are now requesting that all authors identified as 'corresponding author' on published papers create and link their Open Researcher and Contributor Identifier (ORCID) with their account on the Manuscript Tracking System (MTS), prior to acceptance. ORCID helps the scientific community achieve unambiguous attribution of all scholarly contributions. You can create and link your ORCID from the home page of the MTS by clicking on 'Modify my Springer Nature account'. For more information please visit www.springernature.com/orcid.

Best wishes,
Marike

Anne-Marike Schiffer
Editor
Nature Human Behaviour

Reviewer expertise:

Reviewer #1: vision scientist, colour perception

Reviewer #2: vision scientist, circadian rhythm & light

Reviewer #3: vision scientist, circadian rhythm & light

Reviewers' Comments:

Reviewer #1:

Remarks to the Author:

The authors of this registered report propose to study the effect of light varying on the S-L+M axis to the sleep-wake cycle through the supposed effect of this type of cone modulation on the intrinsically photosensitive ganglion cells. The regulatory role of ipRGCs in circadian rhythm and sleep is a much-studied but open question, and warrants further study. The proposed methods seem solid and adequate for addressing the hypotheses laid out in the proposal. The authors have commendably explained their methods and especially the stopping criteria.

There are some methods details that need clarifying; I've listed them below. As a minor detail, the manuscript contains many typesetting errors such as duplicated words which should be corrected before the next submission round. The caption in Box 1 is also cut off.

Introduction

Line 70-72

Please be more specific about what kind of color-opponent system would be appropriate here.

Line 104

Hypothesis about flickering vs constant light: what is this hypothesis based on?

Methods

What is the age range based on? Why not use healthy adults of any age?

Remuneration: do the participants receive any compensation if they withdraw, or if they are not chosen for further study after the adaptation night?

Are subjects allowed to do anything during their time in the lab, such as read a book or listen to the radio?

Reviewer #2:

Remarks to the Author:

The proposed study is exciting, potentially answering important outstanding issues concerning photoreceptor input to ipRGCs in humans. Results will be of broad interest, with clear translational paths.

The proposed study, as designed, will satisfactorily answer the primary research question (on phase shifting). One minor point on the light exposure procedure would be to have staff present to ensure compliance for light exposure, rather than relying on EEG. Having staff present and monitoring compliance will result in better quality data. There are minor concerns on secondary measures. The outcome of interest for the PVT is performance during the light exposure, but it appears a visual test is being proposed. This is potentially problematic for two reasons. First, the visual effects of the light could conceivably impact performance. Second, it is unclear how looking at a computer screen will

impact experimental light exposure. An auditory version of the PVT would be better and has been used to this end (comparing the effects of light exposure on RT previously).

Overall, the logic, rationale, and plausibility of the proposed hypotheses are solid. Methods are sound and feasible. The analysis plan is also acceptable. Methodological detail is sufficient to replicate exactly the proposed experimental procedures and analysis pipeline.

This is an exciting project. There is great confidence that the primary question will be answered using the proposed methodology.

Reviewer #3:

Remarks to the Author:

This is a Registered Report of a study planned to look at the involvement of visual circuit in the stimulation of non-visual functions. The logic behind the project is well laid out and the methodology is sound. Two minor comments:

- 1) There is no mention of whether medications that could impact any of the outcome variables will be excluded, which would be especially important if they are taken as needed.
- 2) The logic behind the use of core body temperature measurement and the requirement of a clear drop in CBT during the adaptation night is not clear.
- 3) Throughout, there is a shift back-and-forth with present and future tense.
- 4) There is inadequate detail provided on the analysis of acute melatonin changes. The ANOVA described in Table 1 implies an independence of the samples, which is untrue. This is true for the other measures that depend on time series.

Author Rebuttal to Initial comments

Response to reviewers

Reviewer #1:

Remarks to the Author:

The authors of this registered report propose to study the effect of light varying on the S-L+M axis to the sleep-wake cycle through the supposed effect of this type of cone modulation on the intrinsically photosensitive ganglion cells. The regulatory role of ipRGCs in circadian rhythm and sleep is a much-studied but open question, and warrants further study. The proposed methods seem solid and adequate for addressing the hypotheses laid out in the proposal. The authors have commendably explained their methods and especially the stopping criteria.

[R1.1] There are some methods details that need clarifying; I've listed them below. As a minor detail, the

manuscript contains many typesetting errors such as duplicated words which should be corrected before the next submission round. The caption in Box 1 is also cut off.

We have fixed the caption in Box 1, which is no longer cut-off. Also, we apologise for the inconveniences due to the typesetting errors and have proofread the manuscript again.

Introduction

[R1.2] Line 70-72: Please be more specific about what kind of color-opponent system would be appropriate here.

We have specified this as follows:

“In general, a colour-opponent system pitting short-wavelength signals against long-wavelength signals is well suited to pick up the spectral changes at dawn and dusk...”

[R1.3] Line 104: Hypothesis about flickering vs constant light: what is this hypothesis based on?

The hypothesis that flickering light has a stronger effect than constant light is based on our standard understanding of how the human visual system adapts to light. At a theoretical level, chromatic adaptation can be modeled using von Kries adaptation. Psychophysically and physiologically, there is overwhelming evidence that such an adaptive effect takes place. We have specified the intellectual reasons for this hypothesis in the text as follows:

Our hypothesis that flickering light will elicit a stronger effect is due to adaptation in cones and postreceptoral mechanisms, for which there is ample psychophysical and physiological evidence³¹⁻³⁷.

We now also explicitly state the reason that yellow-bright flickering light has a stronger effect than blue-dim flickering light:

The hypothesis that yellow-bright flickering stimuli have a stronger effect than blue-dim stimuli is based on the previously discussed animal study¹.

Methods

[R1.4] What is the age range based on? Why not use healthy adults of any age?

Age affects pre-receptoral filtering in front of the photoreceptor receptors, thereby changing their spectral sensitivity. This may lead to small but insignificant uncertainty in the photoreceptor isolation. Since we are testing participants in pairs, a large age range would increase this uncertainty and therefore increase undesired variability. We therefore restrict the age range to minimise any of these effects. We have clarified this for transparency in the section, which now reads as follows:

Healthy, young male and female participants (age range 18-35 years; age range restricted to minimise age-related inter-observer variability in pre-receptoral filtering^{43,44}) will be recruited, with the goal of including an equal number of male and female participants in the study.

[R1.5] Remuneration: do the participants receive any compensation if they withdraw, or if they are not chosen for further study after the adaptation night?

We thank the reviewer for this question. Indeed, your ethical committee requires that participants receive remuneration for every test condition including the adaptation night. To reflect the increasing value of the dataset, the total and maximal remuneration of CHF1000 is composed of CHF50 for the completion of the adaptation night, CHF150 for each of the three test conditions, and a bonus of CHF500 upon completion of the study. Please note that, although the ethical committee will have a final say on the proposed scheme, we expect this to be accepted given previous projects with similar remuneration schemes in our lab.

To keep the manuscript concise, we have decided to not detail the remuneration scheme in the manuscript and hope the reviewer is fine with this.

[R1.6] Are subjects allowed to do anything during their time in the lab, such as read a book or listen to the radio?

We have clarified this as follows:

Throughout the repeated laboratory visits, participants will have no knowledge of external time and not be able to use their phones (except in emergencies) or laptops. They will be able to read magazines, books or other material (e-readers may be allowed if the corneal illuminance is <8 lux). Small gaming devices such as GameBoys (monochrome display), or table-top and card games will be provided. Participants will be able to listen to podcasts, provided they have a device that does not tell the time.

Reviewer #2:

Remarks to the Author:

The proposed study is exciting, potentially answering important outstanding issues concerning photoreceptor input to ipRGCs in humans. Results will be of broad interest, with clear translational paths.

The proposed study, as designed, will satisfactorily answer the primary research question (on phase shifting).

[R2.1] One minor point on the light exposure procedure would be to have staff present to ensure compliance for light exposure, rather than relying on EEG. Having staff present and monitoring compliance will result in better quality data. There are minor concerns on secondary measures.

Thank you for raising this concern. We will ensure that participants are keeping their eyes open and face the light stimulus using an infrared camera system that we monitor from the study suite anteroom. We have clarified this as follows:

Throughout light exposure, participant compliance will be monitored using infrared cameras pointed at the participants.

[R2.2] The outcome of interest for the PVT is performance during the light exposure, but it appears a visual test is being proposed. This is potentially problematic for two reasons. First, the visual effects of the light could conceivably impact performance. Second, it is unclear how looking at a computer screen will impact experimental light exposure. An auditory version of the PVT would be better and has been used to this end (comparing the effects of light exposure on RT previously).

We thank the reviewer for this comment. Indeed, we were planning to use an auditory PVT and have clarified this in the manuscript.

Overall, the logic, rationale, and plausibility of the proposed hypotheses are solid. Methods are sound and feasible. The analysis plan is also acceptable. Methodological detail is sufficient to replicate exactly the proposed experimental procedures and analysis pipeline.

This is an exciting project. There is great confidence that the primary question will be answered using the proposed methodology.

Reviewer #3:

Remarks to the Author:

This is a Registered Report of a study planned to look at the involvement of visual circuit in the stimulation of non-visual functions. The logic behind the project is well laid out and the methodology is sound. Two minor comments:

[R3.1]

1) There is no mention of whether medications that could impact any of the outcome variables will be excluded, which would be especially important if they are taken as needed.

We agree with the reviewer that individuals who take medication impacting on visual, neuroendocrine, sleep, and circadian physiology (determined by study physician) should be excluded. We have now clarified this in the text:

Participants will be excluded if they are pregnant (self-report), suffer from chronic or debilitating medical conditions (physical examination by the study physician), use medications impacting on visual, neuroendocrine, sleep, and circadian physiology (determined by study physician), use drugs (verified with a urine multi-drug screen during each experimental night; nal von minden GmbH, Moers, Germany), have performed shift work <3 months prior to beginning of the study, travelled across more than two time zones <1 month prior to beginning of the study, are characterised by an extreme chronotype according to the Munich Chronotype Questionnaire (MCTQ)⁴⁵ (exclude ≤ 2 or ≥ 7 , include > 2 and < 7), have extremely short or long sleep duration (subjective sleep duration on workdays outside 6-10 hours according to the MCTQ), have a sleep efficiency (i.e., SEFF = total sleep time / time in bed; visual scoring) <70% during the adaptation night, any indicators of a sleep disorder (self-report or during the adaptation night), or have photosensitive epilepsy.

[R3.2]

2) The logic behind the use of core body temperature measurement and the requirement of a clear drop in CBT during the adaptation night is not clear.

We thank the reviewer for this hint. We had included the CBT measurements to make sure participants have a detectable circadian variation in CBT and that light exposure would be timed correctly to elicit an effect. This can be an issue especially with later chronotypes entrained to an early bed time, wherefore DLMO occurs too late with regard to the timing of the light exposure.

Unfortunately, melatonin analyses and the detection of DLMO is not viable as the assays cannot be done within one week (i.e., the time between the adaptation night and the first experimental condition). However, we agree that, especially given the noisiness of the body temperature data (i.e., CBT temperature is posture-dependent), this may not be optimal either. Furthermore, the 'risk' of light exposure taking place too early, seems small in this study as light exposure only takes place between 30 and 90 minutes after habitual bed time. We have therefore decided to take the risk and post hoc exclude data from participants in whom the light exposure did not take place during the rising arm of the melatonin curve.

[R3.3]

3) Throughout, there is a shift back-and-forth with present and future tense.

We apologise and have fixed this in the revision. We now use past tense when we talk about previous research and otherwise use future tense.

[R3.4]

4) There is inadequate detail provided on the analysis of acute melatonin changes. The ANOVA described in Table 1 implies an independence of the samples, which is untrue. This is true for the other measures that depend on time series.

We thank the reviewer for this important remark and hope we were able to clarify this in the updated version of Table 1, which has become the “design table”. Here, we now explicitly state that we will calculate repeated-measures ANOVAs and we name the factors explicitly. The ANOVAs all have one (i.e., where there is only one measurement per condition, e.g. DLMO) to two (i.e., where data are collected repeatedly each evening, e.g. time series of melatonin concentrations) within-subject factors. In repeated-measures ANOVAs, the design takes into account the correlations between sequential samples. We hope this sufficiently answers the reviewer’s remark, otherwise we would be happy to receive further details about her/his concerns.

Decision Letter, first revision:

12th June 2020

Dear Manuel,

Thank you for submitting your revised manuscript, "Effects of calibrated blue-yellow (-S+[L+M], +S-[L+M]) changes in light on the human circadian clock".

I can see that you've provided detailed replies to the referees' comments, and the provision of the code is much appreciated. However, before we can send your work back to the referees, we ask you to undertake an additional change to the manuscript. In detail, we ask you to include a visualization (or description) of the simulation of power estimates in your manuscript. While the provision of code increases robustness and transparency for referees and potential future readers who are versed in R programming, we also need this information to feature centrally in your RR, so that any reader can assess this aspect of your protocol.

I am sorry that I didn't make this requirement explicit in my earlier communication.

We shall hope to receive your revised version as soon as possible.

Please use the link below to submit a suitably revised manuscript and updated response to referees when they are ready.

[REDACTED]

Best wishes
Marika

Marika Schiffer, PhD
Senior Editor
Nature Human Behaviour

Decision Letter, second revision:

9th July 2020

Dear Manuel,

Thank you once again for your manuscript, entitled "Effects of calibrated blue-yellow ($-S+[L+M]$, $+S-[L+M]$) changes in light on the human circadian clock," and for your patience during the peer review process.

Your manuscript has now been evaluated by the same 3 reviewers as before. Their comments are included at the end of this letter. The reviewers' feedback is positive, but we require one last revision of your manuscript before we can make the final decision on Stage 1 in principle acceptance.

Because no changes to your Introduction are permitted following Stage 1 in principle acceptance, we ask you to address Reviewer #1's concern now.

For the work to align with our style and principles regarding authors' presentation of their own work, we further ask you to:

- a) Turn Box 1 into Table 2 (there are no Boxes in research articles; Table 1 needs to be the Design table).
- b) Remove the paragraph "Added value of this study" from what is now Box 1. The added value of your Registered Report is self-evident, and, if anything, can be highlighted at the time you write the Discussion section.
- c) Remove the paragraph "Importance and relevance." from the Introduction. Judging importance and relevance should be left to readers.

In sum, we invite you to revise your Stage 1 Registered Report taking into account reviewer and editor

comments.

* Include a "Response to reviewers" document detailing, point-by-point, how you addressed each referee comment. If no action was taken to address a point, you must provide a compelling argument. This response will be sent back to the reviewers along with the revised manuscript.

* In your cover letter, please include the following information:

- An anticipated timeline for completing the study if your Stage 1 submission is accepted in principle.
- A statement confirming that you agree to share your raw data, any digital study materials, computer code (if relevant), and laboratory log for all eventually published results.
- A statement confirming that, following Stage 1 in principle acceptance, you agree to register your approved protocol on the Open Science Framework (<https://osf.io/>) or other recognised repository, either publicly or under private embargo, until submission of the Stage 2 manuscript.
- A statement confirming that if you later withdraw your paper, you agree to the Journal publishing a short summary of the pre-registered study under a section Withdrawn Registrations.

[REDACTED]

We hope to receive your revised manuscript within four to eight weeks. If you cannot send it within this time, please let us know. We will be happy to consider your revision so long as nothing similar has been accepted for publication at Nature Human Behaviour or published elsewhere.

Nature Human Behaviour is committed to improving transparency in authorship. As part of our efforts in this direction, we are now requesting that all authors identified as 'corresponding author' on published papers create and link their Open Researcher and Contributor Identifier (ORCID) with their account on the Manuscript Tracking System (MTS), prior to acceptance. ORCID helps the scientific community achieve unambiguous attribution of all scholarly contributions. You can create and link your ORCID from the home page of the MTS by clicking on 'Modify my Springer Nature account'. For more information please visit please visit www.springernature.com/orcid.

Best wishes
Marike

Marika Schiffer, PhD
Senior Editor
Nature Human Behaviour

Reviewer expertise:

Reviewers' Comments:

Reviewer #1:

Remarks to the Author:

I think that all of the comments and criticisms have been addressed, but I have one minor issue that you might transmit to the authors before possible acceptance. This refers to my comment that is marked as R1.3 in the rebuttal. It is still not clear from the rewritten paragraph why flickering light should have a stronger effect. Most chromatic adaptation studies have been conducted with static or slowly varying lights (see e.g. Rinner and Gegenfurtner 2000), so there is no logical connection between chromatic adaptation and the effectiveness of flickering light based on the description provided by the authors.

I don't see this as an impediment to Stage 1 acceptance but wish that this would be satisfactorily clarified in the eventual publication.

Reviewer #2:

Remarks to the Author:

The authors have satisfactorily addressed all concerns.

Reviewer #3:

Remarks to the Author:

The authors have adequately addressed my concerns. Good luck!

Author Rebuttal, second revision:

Response to reviewers

Reviewer #1:

[R1.1] I think that all of the comments and criticisms have been addressed, but I have one minor issue that you might transmit to the authors before possible acceptance. This refers to my comment that is marked as R1.3 in the rebuttal. It is still not clear from the rewritten paragraph why flickering light should have a stronger effect. Most chromatic adaptation studies have been conducted with static or slowly varying lights (see e.g. Rinner and Gegenfurtner 2000), so there is no logical connection between chromatic adaptation and the effectiveness of flickering light based on the description provided by the authors.

I don't see this as an impediment to Stage 1 acceptance but wish that this would be satisfactorily clarified in the eventual publication.

We understand the reviewer's point, and wish to clarify it. Under continuous light, cone responses typically adapt. By providing a flickering stimulus, we prevent adaptation to the continuous light source by providing a time-varying input into the visual system. In addition this argument from first principles, we also now cite evidence from a chronobiology study that examined flashing lights, which have shown to exert a stronger influence than continuous light. The rephrased paragraph now states:

Our hypothesis that flickering light will elicit a stronger effect is motivated by the psychophysical and physiological evidence²⁷⁻³³ cones and postreceptoral mechanisms adapt under constant conditions, as well as a recent study showing that flashing light causes a stronger phase shift than continuous light under certain conditions³⁴.

General comments

In addition to addressing the reviewer comments, we also went through the manuscript again carefully, correcting minor inconsistencies and typographical errors.

Decision Letter, third revision:

23rd July 2020

Dear Manuel,

Thank you once again for your manuscript, entitled "Effects of calibrated blue-yellow (-S+[L+M], +S-[L+M]) changes in light on the human circadian clock," and for your patience during the peer review process.

We have read and discussed all the changes you have made to your work, including in response to the Reviewers' concern. I'm pleased to say that your responses to all of these issues are satisfactory.

However, in-principle acceptance of a Stage 1 Registered Report is conditional on the study having received ethics approval. You currently state in the manuscript "Approval for this study is being sought from the local ethics commission". No cover letter has been attached to the current revision to clarify this point.

We cannot proceed to Stage 1 in-principle acceptance until you have been granted ethics approval, which must be clearly stated in the manuscript. We therefore ask you to resubmit your manuscript once this formal requirement is addressed.

* In your cover letter, please include the following information:

- An anticipated timeline for completing the study if your Stage 1 submission is accepted in principle.
- A statement confirming that you agree to share your raw data, any digital study materials, computer code (if relevant), and laboratory log for all eventually published results.
- A statement confirming that, following Stage 1 in principle acceptance, you agree to register your approved protocol on the Open Science Framework (<https://osf.io/>) or other recognised repository, either publicly or under private embargo, until submission of the Stage 2 manuscript.
- A statement confirming that if you later withdraw your paper, you agree to the Journal publishing a short summary of the pre-registered study under a section Withdrawn Registrations.

[REDACTED]

If you cannot obtain ethics approval over the course of the next two months, please let us know. We will be happy to consider your revision so long as nothing similar has been accepted for publication at Nature Human Behaviour or published elsewhere.

Nature Human Behaviour is committed to improving transparency in authorship. As part of our efforts in this direction, we are now requesting that all authors identified as 'corresponding author' on

published papers create and link their Open Researcher and Contributor Identifier (ORCID) with their account on the Manuscript Tracking System (MTS), prior to acceptance. ORCID helps the scientific community achieve unambiguous attribution of all scholarly contributions. You can create and link your ORCID from the home page of the MTS by clicking on 'Modify my Springer Nature account'. For more information please visit www.springernature.com/orcid.

Best wishes

Marike

Marike Schiffer, PhD
Senior Editor
Nature Human Behaviour

Decision Letter, fourth revision:

8th September 2020

Dear Manuel,

Thank you once again for submitting your revised Stage 1 Registered Report, entitled "Effects of calibrated blue-yellow (-S+[L+M], +S-[L+M]) changes in light on the human circadian clock." Everything is in order and I am delighted to say that we can offer acceptance in principle. You may progress to Stage 2 and complete the study as approved.

As you know, a condition of in-principle-acceptance is that the authors agree to deposit their Stage 1 accepted protocol in a repository, either publicly or under embargo until Stage 2 acceptance and publication. We are very keen to showcase our in-principle accepted protocols, so that our readers, reviewers, and potential authors can gain insight into the requirements of the format as well as an idea of the types of projects that are suitable for publication in Nature Human Behaviour. We have set up a space on figshare (https://springernature.figshare.com/registered-reports_NHB) to host all of our in-principle accepted protocols, which can either be made public or kept under embargo until Stage 2 acceptance (depending on author preference). This gives you the opportunity to have your work publicly associated with Nature Human Behaviour, and of course we will be very pleased to showcase your report if you agree to share it publicly.

Depositing the work on our figshare space does not preclude deposition of your Stage 1 protocol on other depositories – your protocol can also be posted on OSF, Dataverse, Dryad or any other public

repository of your choice. You also do not need to do anything – if you agree with posting your protocol on our figshare space, we will upload your protocol on your behalf and either set it public or place it under embargo, depending on your choice. Your protocol will be licensed under a CC BY license (Creative Commons Attribution 4.0 International License). The CC BY license allows for maximum dissemination and re-use of open access materials and is preferred by many research funding bodies. Under this license users are free to share (copy, distribute and transmit) and remix (adapt) the contribution including for commercial purposes, providing they attribute the contribution in the manner specified by the author or licensor (read full legal code: <http://creativecommons.org/licenses/by/4.0/legalcode>) Please note that any use of <https://springernature.figshare.com> will be subject to the Figshare terms of use. Figshare has the right to enforce these terms and conditions where applicable. Use of third party services and sites will be subject to the relevant terms of use and will apply if we act on your behalf in this regard. Do let me know if you would like to take up this option or if you have any questions regarding the protocol deposition requirement.

Following completion of your study, we invite you to resubmit your paper for peer review as a Stage 2 Registered Report. Please note that your manuscript can still be rejected for publication at Stage 2 if the Editors consider any of the following to hold:

- The results were unable to test the authors' proposed hypotheses by failing to meet the approved outcome-neutral criteria
- The authors altered the Introduction, rationale, or hypotheses, as approved in the Stage 1 submission
- The authors failed to adhere closely to the registered experimental procedures
- Any post hoc (unregistered) analyses were either unjustified, insufficiently caveated, or overly dominant in shaping the authors' conclusions
- The authors' conclusions were not justified given the data obtained

We encourage you to read the complete guidelines for authors concerning Stage 2 submissions at <https://www.nature.com/nathumbehav/registeredreports>. Please especially note the requirements for protocol deposition, data sharing, and that withdrawing your manuscript will result in publication of a Retracted Registration.

In recognition of the time and expertise our reviewers provide to Nature Human Behaviour's editorial process, we would like to formally acknowledge their contribution to the external peer review of your manuscript entitled "Effects of calibrated blue-yellow (-S+[L+M], +S-[L+M]) changes in light on the human circadian clock". For those reviewers who give their assent, we will be publishing their names alongside the published article.

When you are ready, please use the following link to access your home page and submit your Stage 2 Registered Report:

[REDACTED]

*This url links to your confidential homepage and associated information about manuscripts you may have submitted or be reviewing for us. If you wish to forward this e-mail to co-authors, please delete this link to your homepage first.

We expect your Stage 2 Registered Report to be submitted by the date specified in your latest cover letter. If unforeseen circumstances prevent submission by that date, please contact us as soon as possible to discuss any changes to the submission time-frame.

Thank you again for offering us this work and we look forward to receiving your Stage 2 Registered Report.

Best wishes,
Marike

Marike Schiffer, PhD
Senior Editor
Nature Human Behaviour

Decision Letter, fifth revision:

Dear Dr Blume,

Thank you for submitting your Stage 2 Registered Report, entitled "Effects of calibrated blue-yellow (-S+[L+M], +S-[L+M]) changes in light on the human circadian clock," and for your patience during the re-review process.

We are very pleased to see that the project has now been completed and the Stage 2 report is ready.

We will shortly send the Stage 2 manuscript to the peer reviewers. However, before we do this, we have noticed that you have made a significant number of changes to the Introduction and Methods sections of the manuscript, compared to the final Stage 1 report.

An important feature of Registered Reports is that the Introduction and Methods should not change at Stage 2 except to change grammatical tense (i.e "we will" to "we did" is allowed). All other changes should be reverted.

In cases where the actual methods used differ from the preregistered Stage 1 methods, you should not edit the Methods text, but instead you should add an additional section to the end of the Methods entitled Deviations from Protocol, listing all of the changes and why these occurred.

Many thanks for making these required changes. We hope to hear from you within 2 weeks; please let us know if the revision process is likely to take longer.

[REDACTED]

With best regards,
Jamie

Dr Jamie Horder
Senior Editor
Nature Human Behaviour

Decision Letter, sixth revision:

26th September 2023

Dear Dr Blume,

Thank you once again for submitting your Stage 2 Registered Report, entitled "Effects of calibrated blue-yellow (-S+[L+M], +S-[L+M]) changes in light on the human circadian clock," and for your patience during the re-review process.

Your manuscript has now been evaluated by Reviewer #2 from the previous rounds of review. Unfortunately neither of the other reviewers were available to re-review, so we recruited a new Reviewer #4 to provide additional comments at Stage 2. The reviewer comments are below.

In the light of our reviewers' advice, we are very pleased to inform you that we will be able accept your Stage 2 manuscript, pending revisions to address reviewer comments and editorial requests.

To assist you with finalizing your manuscript for publication, I attach a checklist that lists all of our editorial requests and formatting requirements.

Please attend to *every item* in the checklist and upload a copy of the completed checklist with your submission. I also mention here a few points that are frequently missed and can cause delays:

- 1) Ensure that all corresponding authors have linked their ORCID to their account on our online manuscript handling system. This is very frequently missed and invariably causes delays in formal acceptance.
- 2) Ensure that you provide all of the materials requested in the attached checklist and below with your final submission.

Nature Human Behaviour offers a transparent peer review option for new original research

manuscripts submitted from 1st December 2019. We encourage increased transparency in peer review by publishing the reviewer comments, author rebuttal letters and editorial decision letters if the authors agree. Such peer review material is made available as a supplementary peer review file. **Please state in the cover letter 'I wish to participate in transparent peer review' if you want to opt in, or 'I do not wish to participate in transparent peer review' if you don't.** Failure to state your preference will result in delays in accepting your manuscript for publication.

Please note: we allow redactions to authors' rebuttal and reviewer comments in the interest of confidentiality. If you are concerned about the release of confidential data, please let us know specifically what information you would like to have removed. Please note that we cannot incorporate redactions for any other reasons. Reviewer names will be published in the peer review files if the reviewer signed the comments to authors, or if reviewers explicitly agree to release their name. For more information, please refer to our FAQ page.

We hope to hear from you within 3 weeks; please let us know if the revision process is likely to take longer.

To submit your revised manuscript, you will need to provide the following:

- Cover letter
- Manuscript text (not including the figures) in .docx or .tex format
- Individual figure files (one figure per file)
- Extended Data & Supplementary Information, as instructed
- Reporting summary
- Editorial policy checklist
- Third-party rights table (if applicable)
- Suggestions for cover illustrations (if desired)

Forms:

Nature Human Behaviour has now transitioned to a unified Rights Collection system which will allow our Author Services team to quickly and easily collect the rights and permissions required to publish your work. Once your paper is accepted, you will receive an email in approximately 10 business days providing you with a link to complete the grant of rights. If you choose to publish Open Access, our Author Services team will also be in touch at that time regarding any additional information that may be required to arrange payment for your article.

[REDACTED]

With best regards,
Jamie

Dr Jamie Horder
Senior Editor
Nature Human Behaviour

Reviewer #2:

Remarks to the Author:

In an impressively designed and executed study, the authors examine the influence on color-sensitive cones on the circadian response to light, something known to be largely dependent on the photopigment melanopsin. The degree to which color-sensitive cones contribute, especially in the early stages of light exposure in humans, is an issue of high importance in understanding the effects of light on the circadian system. There were several deviations noted, but none of these deviations should have negatively impacted the results and most would have enhanced the study. This is a laudable execution that clarifies an important issue.

I have only one important criticism that should be addressed. Though it is touched upon in the Discussion that the results are contrary to the recent St Hilaire et al PNAS paper, what is discussed amounts to little more than the fact that there is a difference between the results/conclusions. I think that more must be done to explain the differences. The St Hilaire study concludes that the response involves a combination of ipRGCs, S-cone, and L+M cone responses. However, this conclusion seemingly hinges on small variations in the data values. For example, the slightly lower value at 480nm seems to be solely responsible for concluding any role for S-cones in the response. This is a data point with a relatively undersized sample (n=11-12). A more parsimonious explanation might therefore be sampling variation, rather than the involvement of an additional photoreceptive pathway. This is especially the case considering the results of the current study, as the 1-h exposure should have been sensitive enough to pick up differential effects (as pointed out by the authors). Simply noting the difference isn't enough, especially as there is good reason to question the other findings. I understand a reluctance to doubt the conclusions of others, but this key discrepancy needs a deeper discussion. I do not consider the evidence equivocal, nor should the authors.

Reviewer #4:

Remarks to the Author:

In this manuscript, Blume and colleagues assess the impact of modulating cones signals across the 'blue'-'yellow' (S vs. M+L) axis on circadian and related responses. Specifically the use within subjects comparisons on the effects of 1h evening exposure to a relatively bright 'white' control light with stimuli that alternate (at 1 Hz) between this control and a 'yellow' or 'blue' spectra of equivalent melanopic illuminance (providing ~25% antiphase contrast steps for S and [L+M] cones). Endpoints are circadian phase delay (as assessed by melatonin DLMO), acute melatonin suppression, subjective sleepiness, PVT, and sleep.

It is relatively well accepted that melanopsin provides the dominant photoreceptive influence on circadian and associated non-image-forming responses under most circumstances. Nonetheless there have been reports in past human studies (as well as findings in animals) that have suggested cones

might also influence such responses under some conditions, including as a result of the colour information they can provide. Such effects, should they occur, have important practical applications and the potential to substantially influence human lighting design and guidance. Accordingly it is very important that such actions be tested in appropriately-designed studies. I therefore commend the Authors for undertaking this work, which is sufficiently well-controlled that the lack of any overt difference between the different stimuli in key outcome measures tested here (eg circadian phase-resetting) provides meaningful information. Indeed, I consider this an important addition to the literature in identifying situations where cones/colour signals clearly don't exert a physiologically meaningful effect. A caveat to that conclusion, however, is that the parameter space is large and the present study doesn't, to my mind, definitively rule out such effects under other practically relevant circumstances. Indeed the stimuli are arguably rather unnatural compared to typical indoor lighting. I appreciate the study design has already been reviewed and approved (barring some deviations) and that it is impossible to test everything. As such, to the extent that I have some reservations about the study, these should not preclude publication. I do, however think some conclusions and statements in the Discussion should be tempered and some additional considerations added.

The main drawback, as I see it, is the choice to use 1Hz flickering test stimuli rather than static 'coloured' lighting. Before my specific suggestions regarding changes/additions to the manuscript, the rationale that opinion is:

A) 1Hz flickering stimuli are fundamentally not representative of those people use (or would be likely to use) in their homes in the evening. While I understand the rationale for the choice (elucidated in the manuscript by the authors) I actually wonder whether the inclusion of flicker masks or dilutes any colour effect. Certainly, key human and animal data that has suggested effects of cones/colour on circadian timing (eg Gooley et al 2010, Mouland et al 2019) used static lighting for long durations, not flickering light like this.

B) As it is, the inclusion of flicker here effectively halves the difference between the two test stimuli and control condition (i.e. the time-averaged irradiances are the mean of 'blue'/'control' and 'yellow'/'control'). In this regard, it is also noteworthy that the present study deviated from the agreed protocol in the magnitude of the 'blue'/'yellowness' of the stimuli (~25% rather than 37% cone contrasts). Accordingly the time-averaged a-opic irradiances experienced by subjects under the two test conditions are not hugely divergent (in the order of 0.1 log units for S, M and L cones). This should certainly be enough to produce a perceptible difference in colour but is much smaller than in the previous studies that provide evidence for effects of cones/colour of circadian phase shifts. It would certainly be fair to say that very extreme coloured stimuli (eg the 460 vs. 555nm monochromatic light compared by Gooley et al 2010) are outside the range that is practically relevant for general lighting use. Conversely, the range of colour tested in the present study is still smaller than that which encompasses commonly used domestic lighting. Fortunately, the 'blue' stimulus is a very cool white (~equivalent to deep twilight even when averaged with the control). By contrast, I believe the 'yellow' stimulus has a colour similar to standard warm white LED/Fluorescent (and approaches D65 when averaged with the background). There are substantially warmer/more 'yellow' lights in common domestic use. Can this study exclude the possibility that if they had used more yellow stimuli (eg resembling candlelight/2700K LED) they might have found a difference?

C) If one assumes that cone contrast responses (colour or achromatic), rather than some steady-state readout of colour, is the important factor - can we be certain these are in any way different between

the two test stimuli? Owing to the inclusion of flicker, under both the 'blue' and 'yellow' test conditions, subjects should experience interleaved increases in L+M cone excitation/decreases in S-cone excitation (stimulus gets more 'yellow') alternating with decreases in L+M cone excitation/increases in S-cone excitation (stimulus get more 'blue') of similar contrast. The only things that really differ between the two stimuli are the midpoints around which these modulations occur (in term of L,M and S-cone irradiance) and the starting phase (which is likely trivial given than the subjects should receive 3600 cycles of each). Is there any direct data which suggests ipRGCs or SCN neurons are likely to show a net difference in activity in response to these two stimuli? Perhaps the inclusion of this contrast cycles actually masks an effect of overall colour?

D) Further to the above, it is worth noting that the melanopic illuminance of all the stimuli used is quite high (~160lx melanopic EDI). Importantly, this should still be on the linear portion of the irradiance response curve but is substantially higher (at least 5X, eg see Cain et al., Sci rep 2020) than is typical for evening home lighting. As the authors already note in the manuscript, the evidence for cone effects on circadian phase resetting reported by Gooley occurred at substantially lower melanopic light intensities. Can the authors therefore exclude the possibility that colour effects might be more apparent at light intensities closer to those typically found in homes in the evening and outdoors around twilight?

Specific Comments:

1) Lines 688:692: 'In a targeted test of our primary hypothesis, there was no conclusive evidence for differential phase-delaying effects of a 1-h nocturnal light exposure (starting 30 min after habitual bedtime) to constant background/control light, blue-dim, and yellow-bright flickering stimuli using typical evening light levels and constant melanopic excitation across light conditions.' – As noted above these are not typical evening light levels. The treatment in abstract ('moderate light levels typical for room illumination') is okay.

2) Lines 693-696: 'Thus, we conclude that, even if there is an effect we have missed, the contribution of a postreceptor channel, where S-cone signals are pitted against a joint L+M signal (i.e. luminance; $S-[L+M]$), is probably not physiologically relevant to the circadian timing system in healthy humans at night under typical light exposure conditions' – I do not agree that the data support this statement as written. As noted above, the inclusion of 1Hz flicker makes the test conditions far from 'typical' nor does the study test the impact of static colours that fully spans the range typical of indoor domestic lighting. The authors should temper the statement and add some content around this point to the Limitations section

3) Somewhere in the discussion, (perhaps somewhere in middle of para 2, which is incidentally rather long and might benefit from being split up, or in the section on Limitations), the Authors should add a section considering the extent to which flickering 'blue' and 'yellow' light might actually be expected to differentially modulate overall ipRGC output/SCN responses and/or whether the inclusion of flicker might have actually masked the effect they aimed to test.

4) Lines 786-788: 'Furthermore, we did not control pupil size, which may have led to small changes in retinal irradiance, but we consider it implausible that this could explain the lack of a conclusive effect.' – The authors may well be correct (and in any case one could argue if a colour effect is only visible under pupil dilation it is not practically relevant anyway). Nonetheless, it would be nice if some more explicit justification could be included. Ideally if the Authors have the direct data or, at least, some

indication of the expected magnitude of such an effect based on the existing literature on chromatic effects on the pupil.

Author Rebuttal, sixth revision:**Reviewer #2**

Remarks to the Author:

In an impressively designed and executed study, the authors examine the influence on color-sensitive cones on the circadian response to light, something known to be largely dependent on the photopigment melanopsin. The degree to which color-sensitive cones contribute, especially in the early stages of light exposure in humans, is an issue of high importance in understanding the effects of light on the circadian system. There were several deviations noted, but none of these deviations should have negatively impacted the results and most would have enhanced the study. This is a laudable execution that clarifies an important issue.

We very much thank the reviewer for the positive feedback and the expressed appreciation.

I have only one important criticism that should be addressed. Though it is touched upon in the Discussion that the results are contrary to the recent St Hilaire et al PNAS paper, what is discussed amounts to little more than the fact that there is a difference between the results/conclusions. I think that more must be done to explain the differences. The St Hilaire study concludes that the response involves a combination of ipRGCs, S-cone, and L+M cone responses. However, this conclusion seemingly hinges on small variations in the data values. For example, the slightly lower value at 480nm seems to be solely responsible for concluding any role for S-cones in the response. This is a data point with a relatively undersized sample (n=11-12). A more parsimonious explanation might therefore be sampling variation, rather than the involvement of an additional photoreceptive pathway. This is especially the case considering the results of the current study, as the 1-h exposure should have been sensitive enough to pick up differential effects (as pointed out by the authors). Simply noting the difference isn't enough, especially as there is good reason to question the other findings. I understand a reluctance to doubt the conclusions of others, but this key discrepancy needs a deeper discussion. I do not consider the evidence equivocal, nor should the authors.

Thank you very much for sharing this perspective and encouraging us to take a more critical perspective. In the revised version of the manuscript, we now make the fact that our findings challenge the conclusions by Gooley and St. Hilaire and colleagues more explicit. Additionally, we have adapted the wording, so their conclusions sound less 'definite'. The paragraph (cf. p. 16, lines 486 et seqq.) now reads:

“Here, the authors concluded that ipRGCs contributed 33%, S-cones 51%, and L+M cones 16% to the phase-shifting effects of the 6.5-h light exposure. Regarding melatonin suppression, during

the first quarter of the 6.5-h light exposure (i.e., 97.5 min), S-cones and L+M cones allegedly substantially contributed to melatonin suppression (51% and 47% contributions of the respective spectral sensitivity curves to the overall sensitivity) while ipRGCs only contributed 2%. As in Gooley et al. (2010)⁵⁰, the contribution of ipRGCs and thus melanopsin significantly increased across time. In the light of these results, we argue that a 1-h exposure specifically targeting the cone-based postreceptor mechanisms should have been sufficient to produce conclusive differential effects between the background condition and the blue-dim or yellow-bright conditions in the present study. Thus, our findings challenge the notion of a strong contribution of cone signals during shorter nocturnal light exposure put forward by Gooley et al.⁵⁰ and St. Hilaire et al.⁵¹. It is important to note that these studies deployed monochromatic lights of different wavelengths and intensities not well suited to stimulate a given photoreceptor class and fit photoreceptor-based spectral sensitivities to the resulting empirical action spectra. Our stimulus design specifically targeted cone-based mechanisms, thereby cleanly isolating the potential contribution of the cones. Future and adequately powered studies using calibrated, photoreceptor-isolating light stimuli and varying irradiance levels, temporal properties and exposure duration are thus needed to settle this question.”

Reviewer #4

Remarks to the Author:

In this manuscript, Blume and colleagues assess the impact of modulating cones signals across the ‘blue’-‘yellow’ (S vs. M+L) axis on circadian and related responses. Specifically the use within subjects comparisons on the effects of 1h evening exposure to a relatively bright ‘white’ control light with stimuli that alternate (at 1 Hz) between this control and a ‘yellow’ or ‘blue’ spectra of equivalent melanopic illuminance (providing ~25% antiphase contrast steps for S and [L+M] cones). Endpoints are circadian phase delay (as assessed by melatonin DLMO), acute melatonin suppression, subjective sleepiness, PVT, and sleep.

It is relatively well accepted that melanopsin provides the dominant photoreceptive influence on circadian and associated non-image-forming responses under most circumstances. Nonetheless there have been reports in past human studies (as well as findings in animals) that have suggested cones might also influence such responses under some conditions, including as a result of the colour information they can provide. Such effects, should they occur, have important practical applications and the potential to substantially influence human lighting design and guidance. Accordingly it is very important that such actions be tested in appropriately-designed studies. I therefore commend the Authors for undertaking this work, which is sufficiently well-controlled that the lack of any overt difference between the different stimuli in key outcome measures tested here (eg circadian phase-resetting) provides meaningful information. Indeed, I consider this an important addition to the literature in identifying situations where cones/colour signals clearly don’t exert a physiologically meaningful effect. A caveat to that conclusion, however, is that the parameter space is large and the present study doesn’t, to my mind, definitively rule out such effects under other practically relevant circumstances. Indeed the stimuli are arguably rather

unnatural compared to typical indoor lighting. I appreciate the study design has already been reviewed and approved (barring some deviations) and that it is impossible to test everything. As such, to the extent that I have some reservations about the study, these should not preclude publication. I do, however think some conclusions and statements in the Discussion should be tempered and some additional considerations added.

We thank the reviewer for the feedback and will address the comments in a point-by-point fashion below. Please note that we have been informed by the editor that we do not need to address comments that solely concern the study design as this was already reviewed and revised during the Stage 1 submission and has been pre-registered. Nevertheless, we tried to provide an answer for all comments as we appreciate the thoughtful feedback and will keep the suggestions in mind for future projects.

The main drawback, as I see it, is the choice to use 1Hz flickering test stimuli rather than static ‘coloured’ lighting. Before my specific suggestions regarding changes/additions to the manuscript, the rationale that opinion is:

A) 1Hz flickering stimuli are fundamentally not representative of those people use (or would be likely to use) in their homes in the evening. While I understand the rationale for the choice (elucidated in the manuscript by the authors) I actually wonder whether the inclusion of flicker masks or dilutes any colour effect. Certainly, key human and animal data that has suggested effects of cones/colour on circadian timing (eg Gooley et al 2010, Mouland et al 2019) used static lighting for long durations, not flickering light like this.

We thank the reviewer for this comment. As pointed out in the manuscript, the 1 Hz flicker was introduced to prevent adaptation of the cones, which typically happens when continuous light is presented (e.g., Webster et al., 2000; Walraven et al., 1990; von Kries, 1970; Rinner et al., 2000). This may be especially relevant when using a, compared to Gooley and Mouland et al.’s studies, rather short light exposure (i.e., one hour). Additionally, temporally modulated light has been shown to generally exert rather large effects compared to constant light (e.g., Lok et al., 2023; Najjar et al., 2016). Thus, we argue it is unlikely that the flicker has diluted the effects.

Additionally, the reviewer rightly hints at flickering light exposure not being representative of domestic lighting, which may limit the generalisability of the findings. We have now included this in the discussion. The paragraph reads (cf. page 18, line 566 et seqq.; also see comment #2 below):

“Finally, the use of a colour-calibrated stimulus flickering at 1 Hz hampers the generalisability of the results to real-life situations. While these conditions were chosen to specifically stimulate the postreceptor S-[L+M] channel while keeping melanopic effects constant, they do not correspond to naturalistic light exposures. Importantly, constant light biased towards either pole of the postreceptor S-[L+M] channel should in any case be less effective than the stimuli used here

due to a likely adaptation of cones and postreceptoral mechanisms under unchanging light conditions^{27-34,70}.”

B) As it is, the inclusion of flicker here effectively halves the difference between the two test stimuli and control condition (i.e. the time-averaged irradiances are the mean of ‘blue’/‘control’ and ‘yellow’/‘control’). In this regard, it is also noteworthy that the present study deviated from the agreed protocol in the magnitude of the ‘blue’/‘yellowness’ of the stimuli (~25% rather than 37% cone contrasts). Accordingly the time-averaged a-opic irradiances experienced by subjects under the two test conditions are not hugely divergent (in the order of 0.1 log units for S, M and L cones). This should certainly be enough to produce a perceptible difference in colour but is much smaller than in the previous studies that provide evidence for effects of cones/colour of circadian phase shifts.

As explained above, we used a flickering stimulus to circumvent adaptation of the cones and postreceptoral mechanisms to a constant stimulus. Thus, we are not convinced that the flicker effectively halved the difference between the two test stimuli. Rather, the stimulation was expected to be more effective than a constant stimulus of equal duration.

It would certainly be fair to say that very extreme coloured stimuli (eg the 460 vs. 555nm monochromatic light compared by Gooley et al 2010) are outside the range that is practically relevant for general lighting use. Conversely, the range of colour tested in the present study is still smaller than that which encompasses commonly used domestic lighting. Fortunately, the ‘blue’ stimulus is a very cool white (~equivalent to deep twilight even when averaged with the control). By contrast, I believe the ‘yellow’ stimulus has a colour similar to standard warm white LED/Fluorescent (and approaches D65 when averaged with the background). There are substantially warmer/more ‘yellow’ lights in common domestic use. Can this study exclude the possibility that if they had used more yellow stimuli (eg resembling candlelight/2700K LED) they might have found a difference?

The colours used here resulted from the constraints imposed by the silent substitution technique, where the aim was to create stimuli that were maximally different regarding their effects on the postreceptoral S-[L+M] channel while keeping melanopic effects constant. More extreme colours likely differ in their excitation of melanopsin besides the cones. This is likely to produce different effects on the circadian system and sleep due to the primary role of ipRGCs in mediating these effects.

We now also discuss this aspect in the discussion section (cf. answer to comment #2 below).

C) If one assumes that cone contrast responses (colour or achromatic), rather than some steady-state readout of colour, is the important factor - can we be certain these are in any way different between the two test stimuli? Owing to the inclusion of flicker, under both the ‘blue’ and ‘yellow’ test conditions, subjects should experience interleaved increases in L+M cone excitation/decreases in S-cone excitation (stimulus gets more ‘yellow’) alternating with decreases in L+M cone excitation/increases in S-cone excitation (stimulus get more ‘blue’) of similar contrast. The only things that really differ between the two stimuli are the midpoints around which these modulations occur (in term of L,M and S-cone irradiance)

and the starting phase (which is likely trivial given than the subjects should receive 3600 cycles of each). Is there any direct data which suggests ipRGCs or SCN neurons are likely to show a net difference in activity in response to these two stimuli? Perhaps the inclusion of this contrast cycles actually masks an effect of overall colour?

Our two stimulus conditions indeed stimulated the blue-yellow channel as described by the reviewer. However, due to the adaption of cones to constant light, it seems unlikely that constant lighting conditions of different L, M and S excitations would have been more likely to produce an effect (please also see response to comment #3 below). Nevertheless, a key question raised by the reviewer is that of temporal contrast at different temporal frequencies in stimulation or hypothesized temporal integration constants. We are unaware of electrophysiological data that have examined this, apart from general contrast-coding neurons in SCN.

D) Further to the above, it is worth noting that the melanopic illuminance of all the stimuli used is quite high (~160lx melanopic EDI). Importantly, this should still be on the linear portion of the irradiance response curve but is substantially higher (at least 5X, eg see Cain et al., Sci rep 2020) than is typical for evening home lighting. As the authors already note in the manuscript, the evidence for cone effects on circadian phase resetting reported by Gooley occurred at substantially lower melanopic light intensities. Can the authors therefore exclude the possibility that colour effects might be more apparent at light intensities closer to those typically found in homes in the evening and outdoors around twilight?

As with any study investigating the effects of light, the results depend on the precise light characteristics including melanopic excitation, the timing of light exposure, or the individual light history. While we had mentioned in the discussion that “future studies will need to evaluate the effects of calibrated colour changes along the blue-yellow axis at different circadian phases, and different mean light levels” (cf. page 17, line 508 et seqq.), we have now also added “melanopic excitation levels”.

Specific Comments:

1) Lines 688:692: ‘In a targeted test of our primary hypothesis, there was no conclusive evidence for differential phase-delaying effects of a 1-h nocturnal light exposure (starting 30 min after habitual bedtime) to constant background/control light, blue-dim, and yellow-bright flickering stimuli using typical evening light levels and constant melanopic excitation across light conditions.’ – As noted above these are not typical evening light levels. The treatment in abstract (‘moderate light levels typical for room illumination’) is okay.

We thank the reviewer for this comment and have adapted the wording in the discussion (cf. page 14 line 431 et seqq.). The sentence now reads:

“In a targeted test of our primary hypothesis, there was no conclusive evidence for differential phase-delaying effects of a 1-h nocturnal light exposure (starting 30 min after habitual bedtime) to constant background/control light, blue-dim, and yellow-bright flickering stimuli using moderate

light levels typical for ~~evening light levels~~ room illumination and constant melanopic excitation across light conditions.”

2) Lines 693-696: ‘Thus, we conclude that, even if there is an effect we have missed, the contribution of a postreceptoral channel, where S-cone signals are pitted against a joint L+M signal (i.e. luminance; S-[L+M]), is probably not physiologically relevant to the circadian timing system in healthy humans at night under typical light exposure conditions’ – I do not agree that the data support this statement as written. As noted above, the inclusion of 1Hz flicker makes the test conditions far from ‘typical’ nor does the study test the impact of static colours that fully spans the range typical of indoor domestic lighting. The authors should temper the statement and add some content around this point to the Limitations section.

Thank you for this critical remark. We have adapted the conclusion and now more specifically speak about “typical room illuminance levels” rather than “typical light exposure conditions”. The sentence now reads (cf. p. 14, line 437 et seqq.):

“Thus, we conclude that, even if there is an effect we have missed, the contribution of a postreceptoral channel, where S-cone signals are pitted against a joint L+M signal (i.e., luminance; S-[L+M]), is probably not physiologically relevant to the circadian timing system in healthy ~~young~~ humans at night under typical ~~light exposure conditions~~ room illuminance levels.”

Additionally, we now discuss the 1 Hz flicker and the fact that the range of colour was smaller than the range commonly encompassed in domestic lighting. Please note that the limited range of colour was not the authors’ choice but resulted from the limitations imposed by the silent substitution technique. The paragraph reads (cf. page 18, line 566 et seqq.; also see answer to comment A above):

“Finally, the use of a colour-calibrated stimulus flickering at 1 Hz hampers the generalisability of the results to real-life situations. While these conditions were chosen to specifically stimulate the postreceptoral S-[L+M] channel while keeping melanopic effects constant, they do not correspond to naturalistic light exposures. Importantly, constant light biased towards either pole of the postreceptoral S-[L+M] channel should in any case be less effective than the stimuli used here due to a likely adaptation of cones and postreceptoral mechanisms under unchanging light conditions^{27-34,70}.”

3) Somewhere in the discussion, (perhaps somewhere in middle of para 2, which is incidentally rather long and might benefit from being split up, or in the section on Limitations), the Authors should add a section considering the extent to which flickering ‘blue’ and ‘yellow’ light might actually be expected to differentially modulate overall ipRGC output/SCN responses and/or whether the inclusion of flicker might have actually masked the effect they aimed to test.

We thank the reviewer for this suggestion. As can be seen in our response to the previous comment, we have now again included the rationale for using flickering rather than constant light in the discussion. As

mentioned in our response to comment C above, the question of temporal contrast at different temporal frequencies in stimulation or hypothesized temporal integration constants remains a key one. However, we are unaware of electrophysiological data that have examined this, apart from general contrast-coding neurons in SCN.

4) Lines 786-788: 'Furthermore, we did not control pupil size, which may have led to small changes in retinal irradiance, but we consider it implausible that this could explain the lack of a conclusive effect.' – The authors may well be correct (and in any case one could argue if a colour effect is only visible under pupil dilation it is not practically relevant anyway). Nonetheless, it would be nice if some more explicit justification could be included. Ideally if the Authors have the direct data or, at least, some indication of the expected magnitude of such an effect based on the existing literature on chromatic effects on the pupil.

To our knowledge, there have been no prior tests of the stimuli we used here in driving pupil responses. We now clarify this as follows (cf. page 18, lines 546 et seqq.):

Furthermore, we did not control pupil size, which may have led to small changes in retinal irradiance, ~~but we consider it implausible that this could explain the lack of a conclusive effect.~~ While it is well known that cone mechanisms contribute to the regulation of pupil size⁶⁰⁻⁶⁴, it is implausible that small variations in retinal irradiance due to differences in pupil size would have counteracted any true photoreceptor-mediated effect. Spitschan et al. (2014)⁶⁴ examined the response of the pupil to flickering stimuli at different temporal frequencies, finding pupillary response amplitudes for 1 Hz stimulation of the L+M channel (50% contrast) around 8%. For a pupil of any size, this corresponds to a difference of roughly 16% in retinal irradiance. We do not expect these differences to play a major role in the effects we see here.

Final Decision Letter:

Dear Dr Blume,

We are pleased to inform you that your Registered Report "Effects of calibrated blue-yellow changes in light on the human circadian clock", has now been accepted for publication in *Nature Human Behaviour*.

Please note that *Nature Human Behaviour* is a Transformative Journal (TJ). Authors may publish their research with us through the traditional subscription access route or make their paper immediately open access through payment of an article-processing charge (APC). Authors will not be required to make a final decision about access to their article until it has been accepted. Find out more about Transformative Journals

With best regards,

Jamie

Dr Jamie Horder
Senior Editor
Nature Human Behaviour